# Sexual signaling pattern correlates with habitat pattern in visually ornamented fishes

Samuel V. Hulse [1✉], Julien P. Renoult[2,3] & Tamra C. Mendelson[1,3]

Sexual signal design is an evolutionary puzzle that has been partially solved by the hypothesis of sensory drive. Framed in signal detection theory, sensory drive posits that the attractiveness of a signal depends on its detectability, measured as contrast with the background. Yet, cognitive scientists have shown that humans prefer images that match the spatial statistics of natural scenes. The explanation is framed in information theory, whereby attractiveness is determined by the efficiency of information processing. Here, we apply this framework to animals, using Fourier analysis to compare the spatial statistics of body patterning in ten species of darters (*Etheostoma* spp.) with those of their respective habitats. We find a significant correlation between the spatial statistics of darter patterns and those of their habitats for males, but not for females. Our results support a sensory drive hypothesis that recognizes efficient information processing as a driving force in signal evolution.

[1] Department of Biological Sciences, University of Maryland, Baltimore County, Baltimore, MD, USA. [2] CEFE, University of Montpellier, CNRS, EPHE, University of Paul-Valery Montpellier, Montpellier, France. [3] These authors contributed equally: Julien P. Renoult, Tamra C. Mendelson. ✉email: hsamuel1@umbc.edu

The diversity of visual patterning across animal species is one of the most striking and enigmatic of evolution's puzzles. While visual patterns often function as camouflage, or evolve through other modes of natural selection, in many cases, they are shaped by sexual selection. Sexual selection is commonly invoked to explain the exaggeration of sexual signals[1]; however, little is known about why particular patterns are selected in some species, while different patterns are selected in others. This question becomes especially perplexing when closely related species exhibit a striking diversity of visual patterns, as in the peacock spiders of Australia, or the birds of paradise of New Guinea.

One of the top candidate hypotheses explaining the evolution of signal design is the sensory drive model of sexual selection. This model posits that the adaptation of perceptual systems to local habitats results in divergent receiver preferences, which in turn generate selection for specific signal features[2–5]. Sensory drive is currently grounded in signal detection theory, such that the attractiveness of a display is a function of its detectability, typically measured as contrast with the background[2,6]. These models are useful for explaining the design of many sexual displays, such as color in freshwater fishes[3], the frequency spectrum of frog calls[7], color contrast in reef fishes[8], and even the push-up display of Anoline lizards[9]. In each of these cases, signals are thought to maximize detectability (contrast) in a given habitat. However, to date, this mechanism of sensory drive has not been as useful for explaining the evolution of intricate visual patterns[6].

Although contrast is also attractive to humans[10], cognitive scientists have discovered the opposite pattern as well, namely that images are attractive when they resemble the spatial statistics of natural scenes. Spatial statistics describe the structural organization of features in an image, and researchers have found that humans both prefer and efficiently process the spatial statistics of natural scenes. For example, visual art has spatial statistics similar to natural scenes, whereas less esthetic images, such as those of laboratory objects (i.e., spectrometers and lab benches) do not[11–14]. Human faces are also deemed more attractive when they resemble the spatial statistics of natural scenes[15,16], and images with natural statistics reduce visual discomfort[17,18]. Interestingly, the spatial statistics of symbols used in human writing also match those of natural scenes, even when symbols with less natural-like statistics are easier to draw[19].

Cognitive scientists hypothesize that people prefer the spatial statistics of natural scenes because our brains have evolved to efficiently process them[20]. Applying information theory to the evolution of sensory systems, Horace Barlow hypothesized that brains adapt to natural statistics by tuning the sensitivity of neurons to the dominant patterns of natural scenes, thereby saving metabolic energy[21]. His "efficient coding theory" has been supported by a large corpus of studies over the past half century, in human and other animals (for reviews, see refs. [22,23]). More

recently, psychologists have shown that efficiently coded stimuli also tend to trigger pleasure and are considered attractive[24–26], thus potentially explaining why humans are attracted to stimuli that match natural spatial statistics.

Recently, Renoult and Mendelson[27] hypothesized that other animals also prefer the spatial statistics of natural scenes, and that this preference could help explain the evolution and diversification of complex animal signal patterns via sensory drive. Their expanded sensory drive framework posits that the neural circuitry underlying sensory perception is tuned to efficiently process habitat-specific features, and that this specialization leads to preferences for particular visual patterns, as might be displayed by a potential mate (also see ref. [28]). However, rather than predicting a contrast between stimulus and background, as in a hypothesis of sensory drive based on signal detection, an efficiency-based model of sensory drive predicts a matching between the patterns of a signal and its background.

In this study, we test the prediction that differences in the spatial statistics of sexually selected phenotypes correlate positively with differences in habitat spatial statistics in a radiation of freshwater fish with complex visual courtship signals (darters, Percidae: *Etheostoma*). Phylogenetic evidence suggests that the most recent common ancestor of darters existed between 30 and 40 million years ago, and darters are now the second most species-rich group of freshwater fish in North America[29,30]. During their breeding season (typically March through May), male darters of most species exhibit species-specific nuptial coloration used in courtship and competition, while females typically remain drab and cryptic. In addition to their striking male color displays, different species of darters exhibit marked variation in patterning (Fig. 1), and mate choice assays have shown that both males and females in some darter species prefer the nuptial coloration and pattern of conspecifics[31–35]. While the most closely related species have similar habitat preferences, distantly related species have divergent habitat preferences that distinguish many sympatric species within a community[36–38]. These distinct habitats could exhibit distinguishable spatial statistics that might drive divergence in pattern preferences and ultimately divergence in the patterns of male sexual signals.

To test for a correspondence between sexual signal and habitat statistics, we capture digital images of ten species of darters and their respective habitats, and use Fourier analysis to characterize and compare their spatial statistics. Fourier analysis is one of the most commonly used methods for analyzing visual images[14,39–42], and indicates how luminance (brightness) contrast is distributed across a range of spatial frequencies. Low and high spatial frequencies correspond to large and small fluctuations in luminance over space (i.e., contrasts), respectively. Large-scale features, such as a uniform sky above a horizon, typically increase the importance (i.e., energy) of the lower spatial frequencies, whereas, fine-scale features, such as grains of sand, increase the

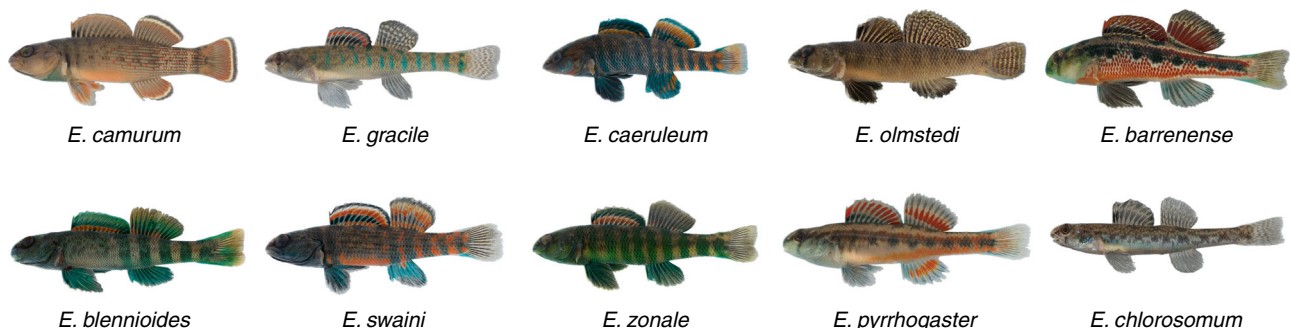

**Fig. 1 Images of males of each species included in this study.** Pictured are males in breeding condition.

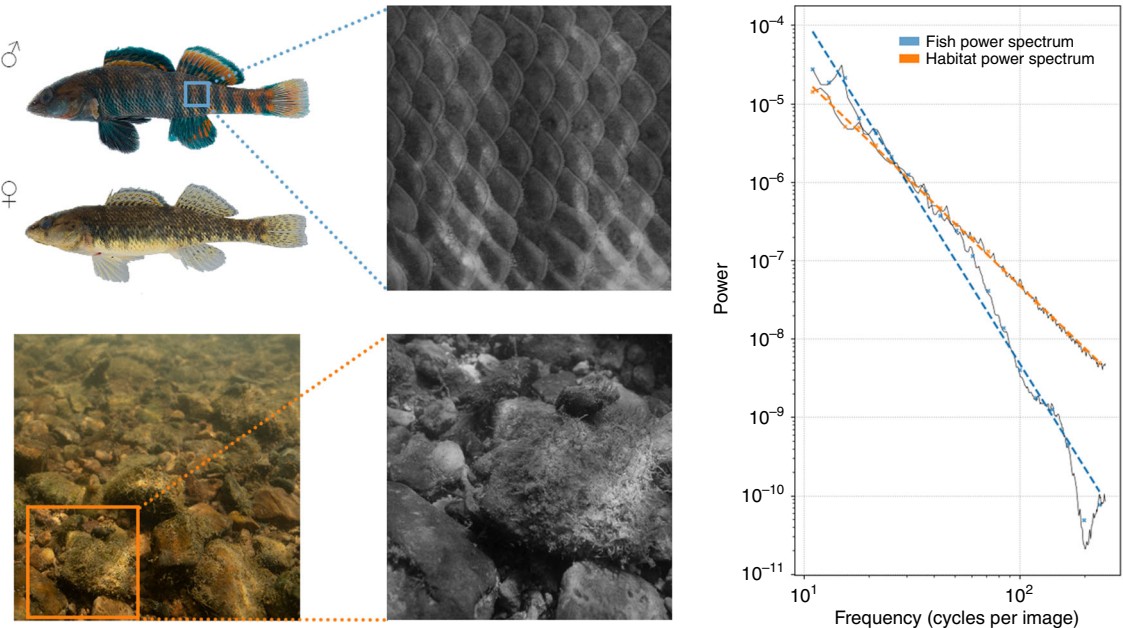

**Fig. 2 Examples of fish and habitat images.** Left column (top to bottom): male and female *E. caeruleum*, as well as an example image of a gravel habitat. Middle column (top to bottom): cropped sections of the original images used for computing the Fourier slope. Third column: Fourier power spectrum for a male *E. caeruleum* (blue) and a gravel habitat image (orange).

energy of higher spatial frequencies. The relationship between spatial frequency and energy is negative for most natural images, meaning that such images are dominated by low-frequency contrasts. When plotted on log–log axes, the relationship is approximately linear, and its slope is referred to as the slope of the Fourier power spectrum (hereafter, Fourier slope, Fig. 2). Widespread use of the Fourier slope in image analysis[12,25,43] is partly due to its biological relevance. Neurons active in the early stages of visual processing in vertebrates are specialized to respond to contrast at specific spatial frequencies. Notably, these specializations correspond to the spatial frequencies that occur with the highest energy in natural scenes[22,44–46]. Stimuli that most closely match the energy distribution of spatial frequencies in natural scenes are the most efficiently processed, as they generate an efficient neurological code that stimulates a small number of highly specialized neurons[23]. The Fourier slope thus has played a central role in demonstrating that the retina and visual cortex of vertebrates are adapted to the spatial statistics of natural scenes[18,46,47].

Here, we investigate an efficiency-based model of sensory drive by testing for a link between the Fourier slope of darter color patterns and their preferred habitats. Analyzing images of ten species of darters distributed across the eastern United States (Supplementary Table 1), as well as images of their habitats, we find a significant correlation between male visual patterns and that of their habitats, which was not present for females. This result supports the hypothesis that sensory drive is acting on the design of sexual signals in darters.

## Results

**Variation in habitat Fourier slope.** The mean Fourier slope of darter habitats was −2.35, which is steeper than the slope of most terrestrial habitats (typically around −2[14,48]). This result is consistent with those of Balboa and Grzywacz[49], who also found that underwater photography of aquatic habitats has distinctively steep slopes, and hypothesized that the optical properties of water blur the visual scene and thereby decrease energy in high-frequency contrasts. We found significant variation in the Fourier

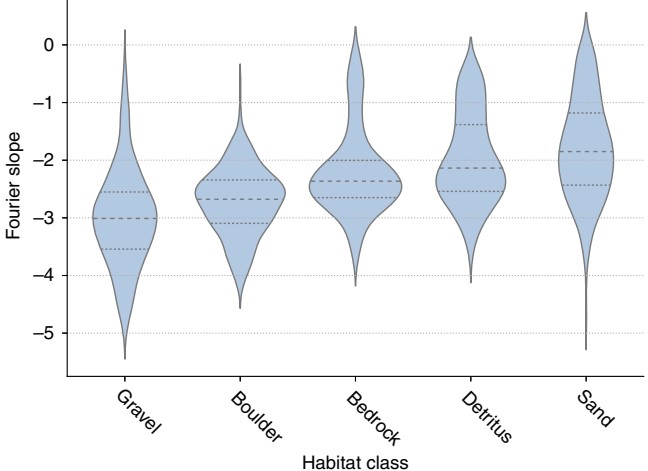

**Fig. 3 Distribution of Fourier slopes for images of five habitat classes.** Dashed lines represent the median for each group, and dotted lines represent the interquartile range. Shaded areas represent the kernel-density estimate for each habitat class. A total of $n = 597$ habitat images were used to represent all habitat types. Each image was subsampled twice for a total sample of $n = 1194$ images.

slope across habitat classes (analysis of variance (ANOVA): $p < 2e-16$, $n = 1194$, $F = 102.3$, $df = 4$, Fig. 3).

**Variation in darter Fourier slope.** We also found significant variation in the Fourier slope between darter species for both males and females (ANOVA, males: $p < 2e-16$, $n = 288$, $F = 17.540$, $df = 9$, females: $p = 5.6e-8$, $n = 262$, $F = 6.256$, $df = 9$, Fig. 4). Overall, males had a shallower slope than females (Student $t$ test: $p = 2.216e-0$, males: $-3.109 \pm 0.318$ SD, females: $-3.275 \pm 0.280$ SD, $df = 550$).

**Habitat–darter correspondence.** We then modeled the Fourier slope of darter patterns as a function of the Fourier slope of their habitat. We determined that the best model included phylogeny, and capture site nested within species as random effects; in the following, we present only the results with these random effects (for alternative models, see Supplementary Note 1, Supplementary Tables 2–5). The relationship between the Fourier slope of visual patterns and that of their corresponding habitat class was significant for males (Bayesian phylogenetic generalized linear model: mean = 0.262, pMCMC = 0.00657, 95% CI: [0.0738–0.4439], here, mean corresponds to effect size, and pMCMC is the Bayesian equivalent of a frequentist $p$ value, Fig. 5a), consistent with a hypothesis of sensory drive based on efficient processing. This relationship was not significant for females (mean = 0.0972, pMCMC = 0.248, 95% CI: [−0.0678 to 0.2722], Fig. 5b) (Supplementary Fig. 1).

Last, we investigated whether males and females differed in how well the slope of their visual patterns matched the slope of their habitat by comparing the slope of each individual fish to the average slope of its habitat class. We found that the deviation of fish slopes from their respective habitat slope was greater for females than for males (mean sex effect = −0.121, pMCMC = 5e

−05, 95% CI: [−0.1631 to −0.0789]). Because one of the primary putative functions of male-signaling patterns is attraction, this result is consistent with the results from cognitive psychology that attractive patterns match the spatial statistics of natural scenes.

## Discussion

Sensory drive posits that animal signals are shaped by the environments in which they are transmitted[2,4,5]. Environmental features affect not only signal transmission but also the sensory and perceptual systems of receivers, which in turn can affect the course of signal evolution. Sensory drive is critical to our understanding of sexual selection because it can explain not only why sexual signals become elaborate, but also why they take their particular form[2,4,5,50]. We found that the habitats occupied by different darter species have different spatial statistics, and that these statistics correlate positively with those of male nuptial patterns, whereas, we found no similar relationship for females. This correspondence supports a sensory drive mechanism for the evolution and diversification of male color patterns in darters.

To date, sensory drive has been rooted primarily in signal detection theory[2,6], and the prevailing prediction is that sexual signals will evolve to contrast with the background against which they are displayed. However, a sensory drive framework rooted in information theory and cognitive science predicts that sexual signals will match the spatial statistics of their backgrounds, based on evidence that natural statistics are efficiently processed and attractive to humans[27]. Our finding that the slope of male color patterns, an attractive sexual signal, was a better match with the slope of their respective habitats than were female color patterns thus supports a sensory drive mechanism based on efficient information coding. This framework may prove useful for understanding the evolution of complex signal design as it incorporates levels of perception beyond detection that might be important for pattern processing[51].

Our finding that the slope of nuptial patterns correlates with the slope of habitats only in males, and that the match between pattern and habitat is better for males than females, makes sense in the context of sexual attraction. If the function of male color patterns is to be attractive to females, then an efficiently processed pattern is advantageous[52]. However, it is puzzling in the context of camouflage, which is the most obvious other reason to expect a correspondence between the visual statistics of animal patterns and their habitats. Indeed, evidence suggests that the Fourier slope is a reliable measure of camouflage. Cuttlefish in camouflage mode match the Fourier slope of their background better than signaling individuals do[53], and damselfish displaying against a background of similar spatial frequency are less attacked by predators[54]. In octopus, rather than matching the entire

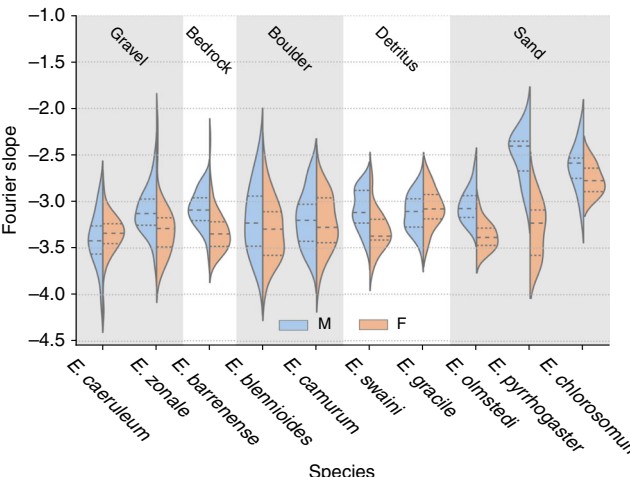

**Fig. 4 Distribution of Fourier slopes for males and females of ten species of *Etheostoma*.** Dashed lines represent the median for each group, and dotted lines represent the interquartile range. Shaded areas represent the kernel-density estimate for each sex within each species. The left (blue) half of each violin plot are values for males (total $n = 288$); the right (orange) half of each violin plot are values for females (total $n = 262$). Species are grouped by habitat type.

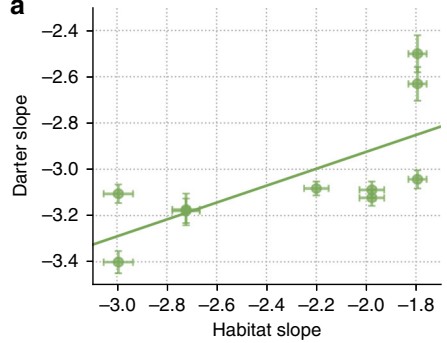
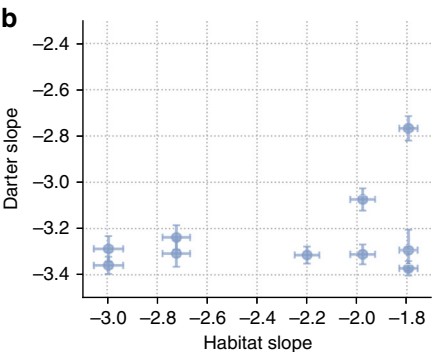

**Fig. 5 Scatterplots comparing mean Fourier slopes of ten species of *Etheostoma* and of their habitats. a** Males ($n = 288$) and **b** females ($n = 262$). For the males, *E. pyrrhogaster* and *E. chlorosomum* have nearly identical values, and appear superimposed on top of each other. This is also the case for *E. camurum* and *E. blennioides*. Error bars represent the standard error.

background, camouflaged individuals match the Fourier slope of background features such as coral or algae patches, also suggesting a camouflage effect of slope matching[55]. Intuitively, then, we would expect females, which are also subject to selection by predation, to follow the same pattern as males. However, a difference between the sexes in darters might be expected for two reasons. First, female patterning in the breeding season is modified by a gravid, white belly that creates low-frequency countershading, which may result in a steeper Fourier slope that departs further from the habitat slope[56]. Lacking pronounced color, females may rely on this detectable countershading to attract males. Several recent studies point to the importance of male choice in this genus[57–60], and future analysis of female patterning in the off season will allow us to test whether female patterns better match their respective habitats when they are not in breeding condition. Second, our analyses considered only luminance (achromatic) contrast, rather than color (chromatic) contrast. The luminance patterning of males may be a camouflaged backdrop against which conspicuous colors are displayed, and this conspicuousness to predators may add increased pressure on males for a camouflaged luminance pattern. Indeed, an effective multicomponent sexual signal might combine a detectable feature like color with an efficiently processed and attractive camouflage pattern. Although we typically model natural and sexual selection acting antagonistically, with sexual signals facing a trade-off between crypsis and conspicuousness, in fact, these forces could align and act synergistically, if camouflage patterns increase attractiveness and promote mate choice.

Our results are consistent with a large body of literature demonstrating that humans prefer images with the spatial statistics of natural scenes. Multiple studies show that visual art is characterized by the spatial statistics of natural scenes, whereas control images are not[52]. For instance, Redies et al.[12] found that artists' portraits of human faces mimic the Fourier slope of natural landscapes more closely than photographs of human faces do, and Graham and Redies[11] summarized evidence that the statistical regularities of artwork are efficiently processed by the human visual system. Other studies demonstrate that people prefer images with natural statistics[11,42]. Spehar et al.[42], for example, showed that human preference for synthetic images peaks at the Fourier slope of natural scenes, and Graham et al.[61] found that preference for landscape images is significantly predicted by their Fourier slope. Images with natural statistics are also less discomforting, with studies showing increasing discomfort correlated with deviation from the Fourier slope of natural scenes[17,18]. Our results, which similarly demonstrate a positive correlation between putatively attractive male signals and natural statistics, therefore extend this literature to non-humans, and suggest that humans may share with other animals a similar biological basis of attraction.

Images with natural statistics are attractive likely because neurons have adapted to process them efficiently. According to Barlow's efficient coding hypothesis[21], sensory systems evolve to create a sparse representation of environmental information, which is thus processed at low energetic cost[62]. A preference for such stimuli is thought to arise because interacting with them is both energetically efficient[52] and effective, that is, it maximizes information transmission and minimizes errors[25,27]. For example, face and object recognition is better (e.g., shorter reaction time and more precise memorization) for images with natural-like statistics[46,63]. The benefits of displaying sexual signals with natural-like statistics also could arise from short-term neurophysiological processes. Studies in humans have shown that matching the slope value of a target stimulus (e.g., a face) with that of the surrounding background has a positive effect on cognitive performance[64], and on ratings of attractiveness[15], which

is likely mediated by sensory competition and/or short-term adaptation. The precise mechanisms linking these neurophysiological processes to behavioral changes remain to be elucidated[64].

Last, our finding that variation in the Fourier slope of darters is consistent with variation in species' preferred habitat may have implications for human esthetics. The Fourier slope has been found to vary between different categories of stimuli (e.g., buildings, natural landscapes, and anthropogenic objects), and between terrestrial and aquatic environments[48,49]; however, the extent to which natural human habitats vary in Fourier slope has not yet been explicitly addressed. Different terrestrial biomes (e.g., tropical forest, desert, and seashore) very likely have different Fourier slopes; thus, quantifying how they covary with works of art and regional esthetic preferences may further contribute to understanding the mechanisms driving aspects of cultural evolution and diversification.

In conclusion, we note that our study is limited in scope to a single spatial statistic in a single taxon and does not directly investigate perceptual processing. In addition, we do not yet have critical behavioral data testing whether the fish indeed prefer natural-like statistics, or neurophysiological data testing whether these patterns are more efficiently processed. Nonetheless, our results support a plausible explanation for how sensory integration beyond stimulus detection can drive signal design through an environmentally mediated process. Through the lens of Fourier analysis, we have provided evidence that male visual signals correspond to the visual statistics of their habitats, suggesting that post-retinal visual processing could explain certain aspects of signal design. Our hypothesis is a novel extension of sensory drive, and our methods provide a new approach for testing the role of sensory drive in the evolution of visual patterns. Because the animal patterns we studied are likely used in mate attraction, our results also support key predictions of cognitive science about the relationship between attractiveness and natural scenes. We suggest that some of those principles extend beyond humans and provide a compelling hypothesis for how a complex trait can evolve in a predictable, environmentally dependent direction.

## Methods

**Darter collection and photography.** We collected males and females of ten darter species from 23 sites distributed across Illinois, Kentucky, Maryland, Missouri, Pennsylvania, and Tennessee (*Etheostoma barrenense*, *Etheostoma blennioides*, *Etheostoma caeruleum*, *Etheostoma camurum*, *Etheostoma chlorosomum*, *Etheostoma gracile*, *Etheostoma olmstedi*, *Etheostoma pyrrhogaster*, *Etheostoma swaini*, and *Etheostoma zonale*; Supplementary Table 1, Supplementary Note 2). These ten species were chosen for inclusion based on their broad phylogenetic distribution and their preference for different classes of habitat: sand (*E. chlorosomum*, *E. olmstedi*, and *E. pyrrhogaster*), boulder (*E. blennioides* and *E. camurum*), gravel (*E. caeruleum* and *E. zonale*), detritus (*E. gracile* and *E. swaini*), and bedrock (*E. barrenense*). Permits for scientific collection were obtained through the Kentucky Department of Fish and Wildlife Resources (Permits SC1711119 and SC1811151), Tennessee Wildlife Resources Agency (Permits 1052 and 1424), Illinois Department of Natural Resources (Permits A17.6089 and A18.6089), Missouri Department of Natural Resources, Louisiana Department of Natural Resources (Permit SCP 183), Mississippi Department of Wildlife, Fisheries, and Parks (Permit 0219184), and Maryland Department of Natural Resources (Permit # SCP201747). At each site, we collected approximately 10 males and 10 females (males: $11.5 \pm 2.2$ SD; females: $10.5 \pm 3.1$ SD), which were subsequently photographed. Darters were caught by kick-seining and brought back to either the Hancock Biological Station in Murray, KY, or University of Maryland Baltimore County. Fish were housed in aerated tanks and photographed within 3 days of capture. Immediately prior to photography, fish were euthanized in MS-222 and then fixed in 10% formalin with fins pinned erect for approximately 10 min. We then clipped the pectoral fin of each fish for an unobstructed image of their body pattern. Fish were then removed from the formalin and placed in an open-topped cylindrical glass photography arena with water at a depth of 2 cm. The arena was surrounded by diffuser paper and illuminated by three Canon 270EX II flashes spaced equidistant around the arena. Each fish was placed in the arena with the camera facing perpendicular to its side. We then photographed each fish using a Canon EOS 5D Mark IV digital camera with a Canon EF 100 mm f/2.8 L macro lens attached. The camera was mounted on a Cognisys Stackshot Extended Macro Rail to enable automated focus stacking. Prior to imaging, the camera's position on the rail was set so that the most

proximate part of the darter was in focus. The macro rail was then set to capture an image every 0.5 mm, until the most distal part of the fish was in focus. During every photography session, we also captured an image of a white balance card in the photography arena. Our animal treatment protocols were approved by UMBC IACUC Protocol no. TM01841518 and comply with all ethical regulations for animal testing and research.

**Habitat photography**. We collected underwater images of habitats at sites where we captured darters using an Ikelite 200DL underwater housing for a Canon EOS 5D Mark IV digital camera equipped with a Sigma 24 mm f/1.4 lens. Each darter species was assigned to a habitat class (sand, gravel, boulder, detritus, or bedrock) based on where darters were observed, as well as on literature describing darter microhabitat preferences[65–67]. Each image was captured by holding our camera near the bottom of the stream and taking pictures facing multiple directions. This was repeated across each field site until the area where we found darters were photographed to the complete extent than conditions allowed, and until each habitat had at least 100 photographs. We also took images of a waterproof white balance card (DGK Color Tools WDKK Waterproof Color Chart) at each site. Every habitat class was represented by a minimum of 100 images representing a minimum of two sites. All images were collected in clear, shallow water on sunny days between 10:00 and 15:00, and when water turbidity was low.

**Image processing**. To retain the high dynamic range and linearity of the RAW image files, we used a custom python script using the library libraw to extract image data from each Canon RAW file. We applied no gamma correction or white balance. In addition, each image was reduced to half size to avoid nonlinearities associated with demosaicing algorithms. For darter images, we then combined each image stack to a single image using Zerene Stacker with the DMap algorithm.

**Image conversion to darter color space**. For all images, RAW files were converted to 16-bit tiff files using the rawpy python API. We converted RAW files to RGB triplets without any spatial interpolation, gamma correction, or white balance to maintain linearity. Images were then converted into a darter color space. The generation of the darter color space was done by first characterizing the sensitivity functions of our camera, and then using known darter visual sensitivities to generate a mapping from camera to darter space[68]. The sensitivity functions for the camera were estimated using a monochromator and a calibrated spectrometer, which ensured that each color channel was linear[69]. To model darter color vision, we generated a dichromatic model using cone sensitivities peaking at 525 and 603 nm (darters lack a short-wavelength-sensitive cone class). Since the cone sensitivities for all species in this study are not currently known, we used the same color vision model for all species. Variation between darter species in cone sensitivities is known to be relatively minor and unlikely to affect the outcome of our analysis[68]. We converted camera space to darter space by minimizing the difference between the camera and the cone model, using a second-order polynomial function of RGB values. We then converted all images (fish and habitat) from color space to luminance space by summing the two color channels. This pooling of color channels closely mimics how vertebrate brains are thought to extract luminance information[70].

For darter images, we cropped out a rectangular region on the flank of each fish directly below the second dorsal fin. From this set of cropped images, we determined the largest square area that would fit in every image, which was found to be $200 \times 200$ pixels. We then randomly sampled each flank image twice with a $200 \times 200$-pixel box.

Habitat images were reduced from their original dimensions of $2251 \times 3372$ to $600 \times 900$ pixels. We then randomly sampled each habitat image twice with a $200 \times 200$-pixel square to match the dimensions of the fish images. Varying box size had no qualitative effect on our results (see Supplementary Note 3, Supplementary Tables 6 and 7). In addition, we ensured that there was no bias in the proportion of images in each habitat class that had a visible horizon (Supplementary Note 4, Supplementary Table 8).

**Fourier analysis**. To compute the slope of the Fourier power spectrum for each image, we followed standard methods in visual psychophysics and empirical esthetics[12,43,71]. We calculated the two-dimensional Fast Fourier Transform with a Kaiser–Bessel window using parameter $\alpha = 2$ to minimize edge artifacts[48]. We then transformed the Fourier space to the power spectrum and estimated the radial average of the power spectrum. To eliminate edge effects and high- frequency noise, we only included spatial frequencies between 10 and 110 cycles per image. Since the Fourier power spectrum has a greater spatial granularity at higher frequencies, we binned ($n = 20$) each power spectrum between 10 and 110 cycles per image. This ensures that our slopes were calculated to give equal weight across the frequency range. We then estimated the slope of the power spectrum using a linear regression on the bin values, using a custom Python script.

To compare the slope of each individual fish to the average slope of its habitat class, we first computed the bin values for each habitat class, and then defined the deviation for an individual fish as the mean-squared error between the fish bin values and the habitat bin values. We then used these deviation values in a

MCMCglmm model (below) to test the effect of sex on deviation of the individual pattern from habitat slopes.

**Statistical analysis**. To ensure that images of species-typical habitat were representative of their class, we pooled images of each habitat class across multiple sites. We first used a one-way ANOVA to test whether the Fourier slope of habitats varied by habitat class. We also used a one-way ANOVA to determine whether the Fourier slope of darter color patterns varied by species. This ANOVA was performed separately for males and females. To compare the Fourier slope of males versus females, we performed a two-tailed $t$ test.

To examine the relationship between the Fourier slope of habitats and that of darter patterns, we used generalized linear mixed models. We computed this model using the R package MCMCglmm[72,73]. MCMCglmm can account for phylogenetic covariance and uses Markov chain Monte Carlo (MCMC) sampling to generate posterior distributions. Our model predicted the value of the Fourier slope of each individual fish based on the slope of its habitat. We included capture site nested within species as a random effect. The phylogenetic tree of the ten studied species was inferred from a previously published molecular phylogeny (accessed via TreeBASE)[30,74]. We used the uninformative Inverse-Wishart prior and a normal prior for the fixed effects (for all effects: prior of belief parameters nu = 0.02 and of variance parameter $V = 1$). We then ran our model for 1,000,000 iterations with a burn-in period of 10,000, thinned every 50 iterations. We visually examined traceplots and checked diagnostics to assess convergence, and estimated the effect of habitat slope (*means* and 95% CI) from its posterior distribution. To evaluate the strength of the phylogenetic signal, we compared the DIC of models with and without the phylogeny (Supplementary Tables 2–4).

We also used MCMCglmm to test whether male and female patterns differed in their deviation from habitat slopes, modeling deviation of each individual as a function of sex, with capture site nested within species and phylogeny included as random effects. We used the same model parameters as above.

**Reporting summary**. Further information on research design is available in the Nature Research Reporting Summary linked to this article.

## Data availability
The computed slopes for every image used can be accessed via github at https://github.com/svhulse/Fourier-Analysis. Any images used in this study are available upon request. The source data used for Figs. 3–5 are provided as a Source Data file. The phylogenetic data used to generate our models can be accessed via TreeBase at https://treebase.org/treebase-web/search/study/trees.html?id=11548.

## Code availability
All code used to compute the slope of the Fourier power spectrum can be accessed via github at https://github.com/svhulse/Fourier-Analysis.

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

## Acknowledgements

We would like to thank Dr. Thomas Cronin for help with camera calibrations, Matthew Dugas, Natalie Roberts, and Rickesh Patel for assistance with field collections, and the Hancock Biological Station for providing a home base for field work and photography. This work was supported by National Science Foundation grant IOS-1708543 and by the CNRS (PICS project n°08302).

## Author contributions

T.C.M. and J.P.R. conceived and designed this study. S.V.H. collected fish with assistance from T.C.M. and J.P.R., and was responsible for all the photography. In addition, S.V.H. wrote all python code and performed all analyses. All authors worked to write and edit the paper.

## Competing interests

The authors declare no competing interests.
