## [Peer Review File · Nature Communications]

Reviewers' Comments:

Reviewer #1:

Remarks to the Author:

This is an exciting finding within the field of visual ecology showing that spatial elements of sexual signals covary with habitat. The data are all about the physical characteristics of signals and background, which makes it puzzling that the interpretation focuses so heavily on the subjective evaluation of signals by receivers. While the connection to empirical aesthetics (a very small field with a Max-Planck Institute) is intriguing, these results fall squarely and elegantly within the realm of visual ecology and psychophysics. I understand the temptation to connect these results to the authors' recent conceptual paper on processing fluency, but I think the connection is far overstated here. The model being presented here is analogous to sensory drive: habitat characteristics drive perceptual biases. Signals are favored because they address these biases. With these data, it is impossible to distinguish between whether signals are favored because they are easier to detect by receivers, versus whether they are processed more pleasurably. I found it surprising that there wasn't more of a connection to psychophysics and what we already know about processing of spatial information. One of the interesting things about spatial frequency is that it is processed at multiple levels of the visual system, starting with receptive fields in the retina. Spatial frequency sensitivity is also famously sensitive to the early visual environment, providing a compelling mechanism for the tuning of frequency sensitivity to the local visual environment. It is possible that these spatial frequencies are more detectable or more attractive because they're parsed more easily in a given environment. But visual systems function by detecting contrast, and matching spatial frequency is a common feature of camouflage and mimicry. Indeed, I think the most parsimonious evolutionary explanation for this pattern is that the spots of male darters have evolved to minimize conspicuousness to color-insensitive fish predators. Colors thereby stand out for conspecific females while predators see a homogeneous background. It is worth revisiting some of the earlier literature on visual ecology, from Fleishman's work on temporal-frequency contrast in courtship displays to Changizi's work on written symbols and topological patterns in natural scenes to Justin Marshall's extensive work on communication and camouflage in reef fish. Rosenthal's Annual Review on visual communication talks about spatial frequency matching in camouflage. Finally, this work presents an exciting analogy to auditory frequency and pattern tuning and habitat/eavesdropper effects on signal/receiver evolution. That literature, grounded in psychophysics and neurophysiology, provides more promising grounds for future research building off this data set than does the interesting, but tenuously related, field of empirical aesthetics. The one substantive comment I would make is to reanalyze the data in terms of spatial frequency contrast and testing for contrast differences across the spatial-frequency spectrum. This will allow you to test whether there are spatial frequencies where darter signals stand out against the background. This will then provide direct predictions for future studies on the spatial-frequency sensitivity of receivers. To summarize, this is an interesting and important study that could make a substantial impact with some scholarly revision and additional analysis.

Reviewer #2:

Remarks to the Author:

I really enjoyed reading this manuscript. In this manuscript, the authors tested the idea that what makes a particular signal attractive is based in what the sensory system is primed to see. This idea has been around for a while in the area of human perception of art and even in the way that letters are formed. The idea is that animals are very efficient at detecting particular patterns that are common in their environment. Things like art (in case of humans) or male secondary sex characters (in the case of the darters at hand) might have elements of their color patterns that reflect this bias.

To test this, the authors took pictures of a number of male and female darter species that are widespread in their placement in the darter phylogeny. They took pictures of males and females below the second dorsal fin. They also took pictures of the environments in which the fish are found. Some are found in rocky areas, others in detritus, others over sandy substrate.

They then performed an analysis involving the fourier transform. I will admit that I do not fully understand this method. I am not fully clear on what the 'slope of the Fourier power spectrum' represents. This should be spelled out a little bit more clearly in the methods.

They then performed Anovas on the slopes. They also examined the relationships between the Flourier slope of habitats and darter color patterns, but for some reason they used an MCMCglmm approach. I don't understand why they didn't use a normal linear regression approach. My issue with the MCMCglmm is that you usually have to specify some type of underlying distribution. We don't know what that is in this case. I looked in the supplemental information but did not see it. The reason I am curious about this is because the confidence limits on line 133 the confidence limits for the slope gets super close to zero (1.5×10^{-17}). The value on this is pretty small (5×10^{-3}), but the upper limit on the confidence limit is 0.0252. This suggests that confidence limit is constrained to not go below zero. If this is the case, then this result needs to be reinterpreted. Again, we need more detail on the distribution of the MCMCglmm.

More generally, I found myself wondering whether matching the energy spectrum (i.e., congruence between the two fourier slopes) should make one attractive or whether it should make one cryptic. If we had found the opposite pattern (i.e., a good match for females, but not for males), then would we be interpreting this in the light of crypsis.

I also had a bit of difficulty in interpreting figures 3 and 4.

My recommendation is that the authors make this paper a bit more reader friendly. Many of the folks reading this will be coming from an ecology and evolution perspective, so they are going to need some hand holding on the fourier analysis section. What is a fourier transform? I have done these in the past, but I would have problems explaining them. Lots of folks will have never done these. So a bit of hand holding in the supplemental materials would be appreciated.

If the authors can fix these issues, then I think that this will make for a very nice paper in Nature Communications.

Reviewer #3:

Remarks to the Author:

The manuscript "NCOMMS-19-24571 - Sexual signals of fish species mimic the spatial statistics of their habitat: evidence for processing bias in animal signal evolution" deals with the similarity of Fourier slopes between male darter fish and their habitat to test a signal drive hypothesis of evolutionary biology. It is a very well-written manuscript and a well-controlled study. However, in the current state the justification of the exact research question and its novelty is not perfectly clear to me. Although I believe (but not know) that it is the first study that uses image processing methods to test a theory on sexual selection, it seems to disregard related literature from psychology, for example using human face stimuli. Therefore, I suggest the revision of the respective paragraphs in the introduction (is indicated below). I am also not contented that (only) fish species are considered in this study. Analyses of other groups of species would be much appreciated to strengthen the drawn conclusions. Furthermore, the result section needs to be revised because in the current state the analyses and their statistical indices are not clear and might be improved. Furthermore, some methodological aspects are not clear in the current state of the manuscript and should be addressed. Additionally, the discussion section doesn't deal much with the actual data presented but rather more general with the hypothesis tested. In my opinion, the current data and its limitations should be discussed in more detail. Based on the more detailed comments below, I suggest a major revision of the manuscript.

It should be noted that I read the manuscript in the order any 'Nature communications' reader would read it (Abstract – Introduction – Results – Discussion – Methods – Supplemental) and, therefore, some of my comments might be overridden after reading later sections (esp. Methods). However, it is an indication that information is missing at a point where it is necessary. So the authors should revise the respective sentences, even when it is described in detail at later sections.

Line 31: The reference is not numbered.

Lines 63ff: "In parallel, other studies have shown that...". Please cite the studies you are referring to.

Line 68: "hypothesize" ◇ hypothesized

Line 71f: Although I agree that in most studies natural scenes were not considered in their diversity, the current sentence seems to claim that no researcher ever thought about different sceneries and habitats. I'd prefer a less absolute wording.

Line 80ff and generally to the research question and hypothesis: For me, it is not clear

whether the current study is the first attempt to test the hypothesis stated here. What is about, for example, the relation of the characteristics of human faces and their assumed natural habitat? In fact, photographs of human faces have different statistical properties than natural scenes (Redies, Hännisch, Blickhan, & Denzler, 2007). Further and more importantly, how does this theory/hypothesis deal with the fact that several different-looking animals live in the same habitat (even when considering only animals that use visual cues for mate attraction)? Additionally, the authors may also refer to other perception studies, e.g. on fore- and background segregation and sensory competition. So far, I also do not understand why the properties of the entire habitat rather than particular (rewarding) elements in it should correlate with the visual cues of the fish. I'd appreciate a more detailed explanation and justification of the research question and hypothesis. However, I should note that I am not very familiar with sensory drive models and related hypothesis. Furthermore, I was surprised that fish species were selected as the target (and the only one). At one point, it should be described whether and how visual perception of fish (in water) differ from other animals (on land) and whether this has an effect on the used measure. To allow a more universal conclusion, the authors may consider analysing data also from other species' groups (preferably from land). Overall, the justification of the research question and hypothesis is not entirely clear and satisfying to me.

Line 106f: The horizon line is represented by high rather than low spatial frequencies. The authors may rather refer to the change in luminance between sky and foreground for example.

Line 111f and elsewhere: The authors may also refer to related work of other groups and research foci, such as work on visual discomfort and Fourier slope (e.g., Juricevic, Land, Wilkins, & Webster, 2010). Furthermore, although I agree with the appropriateness of the Fourier slope as the selected measure, it should be considered to analyse further statistical properties (and if decided against it, it should be briefly stated why the Fourier slope is considered the only relevant measure).

Line 117ff: I don't understand the indications for the statistical tests and therefore I am not able to approve their correctness. If that refers to an ANOVA, the statistical indices should be indicated correctly. Further, the p-values should be indicated as $p < .001$ or similar, rather than " $p = 2e-16$ ". I am also not sure why sex seems to be not included in the ANOVA as a factor. Rather, it seems that two different ANOVAs were calculated for the two sexes. The indications for the statistical tests should be revised and if appropriate the statistical analyses should be redone including all factors in one analysis.

Line 118ff: If "Overall, males had a higher slope than females..." refers to a main effect in the calculated ANOVA, this and the respective statistical indices should be indicated. If not (which might be indicated by "this difference was significant in 5 out of 10 species"), it shouldn't be stated that "overall" males had a higher slope. In general, the description of the results is not precise and understandable.

Line 123: It is not clear what "phylogeny" refers here to. It is also not clear to me, why sex and species is not entered to the model as fixed factors.

Lines 128ff: The authors may reconsider the numbers of decimal places.

Line 129f: "We did not find a strong effect of the..." ◊ I'd prefer: "We found a rather small influence of..." since the authors did find a significant effect.

Line 135: For the sake of completeness, the statistical indices for that alternative model should be stated.

Line 148ff: This paragraph might be moved to the introduction because no current data are discussed here while it rather discusses the research gap the study aims to fill. If the authors prefer to keep it in the discussion, it should be shortened and the current data should be integrated.

Lines 161ff: Again, this and the following paragraph do not discuss the current findings but rather (partly redundantly) describes previous research. They should be moved to the introduction, or shortened and connected to the current data.

Line 191ff and elsewhere: Although I am much aware that it is not good manners to suggest own studies as a reviewer, I'd like to draw the authors' attention to studies on human faces in front of patterns with different Fourier slope backgrounds (e.g., Menzel, Hayn-Leichsenring, Langner, Wiese, & Redies, 2015; Menzel, Hayn-Leichsenring, Redies, Németh, & Kovács, 2017; Wu, Xu, Dayan, & Qian, 2009). I am not asking for being cited but would like to point out that there exist studies on the perception of the similarity of statistical properties of backgrounds (which might be interpreted as 'habitat' in the context of perceiving and evaluating the target in the foreground) and the targets in the foreground. Furthermore, within this context, the authors may also discuss further perceptual mechanisms, such as sensory competition and (short-time) adaptation.

Line 207: Because of its importance to the current study, I'd prefer a reference to original data on the Fourier slope in terrestrial compared to aquatic environments, rather than referring to an overview book. Also for aquatic environments, it should be explicitly stated whether this difference refers to photographs made from above or below the water surface. Further, I am surprised that the absolute numbers of the Fourier slope retrieved for the habitats and fish in the current data set are not discussed in relation to the fish, and to other image categories analysed in previous studies. The slopes of the fish habitats are rather steep compared to those of natural scenes found in previous studies. Also, the slopes of the fish are even steeper (and much likely significantly different from their habitats), which is a bit surprising when looking at the images because I can see a lot of fine detail on the fish surface. I am wondering whether the curve of the power spectrum is really a straight line for the fish. Similar to figure 2, the authors should include example curves for the fish. The authors should also make clear why they think the correlation between fish and habitat slope is more informative than comparing the slopes directly. Relatedly, the authors may revise their title and abstract to adjust the conclusion to their data. Since the conclusion is made based on a correlation and not a direct comparison, in my opinion, one cannot say that the fish "mimic" (title) the statistics of their habitat, and that these

statistics are "compared" (abstract).

Discussion in general: Overall, the current results are not discussed in much detail. I suggest, revising the discussion to reduce redundancy with the introduction and to address the current data in more detail as well as the limitations of the study (e.g., in statistical terms a quite low number of habitats and fish species which are correlated; focus on only one particular group of fish species'; focus only on aquatic animals; focus only on Fourier slope etc.).

Line 271ff: Example images of the cropped and adapted versions of the fish photographs would be very helpful to follow the technical paragraphs and to interpret the Fourier slopes obtained. Please provide at least one example image that was used to measure the Fourier slope.

Line 274: Please add a break/paragraph to separate the description for fish and habitat photographs.

Line 276f: "Since the size of the habitat images is greater than the size of the darter's flank, using a larger box size reduces variability in lower frequency coefficients." \diamond I don't get that sentence. Please also explain why different image size is not a problem for your analysis of Fourier slope (usually it does affect the slope). The "Additionally" at the beginning of the following sentence makes no sense to me, because the previous one didn't explain why different image sizes are no problem.

Line 281ff: Related to the previous comment, I don't understand why the same range of frequencies is considered for habitat and fish images which differed in size. When focusing on only a subset of frequencies for a larger image, information about the lower spatial frequencies are lost and the two slopes are not comparable. This should be explained and possibly changed.

Line 287ff: Okay, I see that the authors tried to match the analyses by adjusting the bins. However, I am not sure whether it solves the problem of differing image sizes. I do not understand why it was worked with different image sizes at all. This should be made more clear. I'd suggest identical image sizes and analysis procedures for both image types. Also, the sentence in line 286f and the following one seem to contradict each other. When only frequencies between 10 and 110 were included, how can the binning include the frequencies 10 to 200? This is not clear.

Line 394ff: As stated above, it is not clear why not all relevant factors are included in one ANOVA.

Line 299ff: I am not a statistician but in my understanding phylogeny should be included as a fixed rather than random factor because one might expect systematic and predictable influences of phylogeny (as for example sex which is usually entered as a fixed factor in such models). Please also explain how this information is entered in the model. So far, I cannot imagine how a phylogenetic tree can be entered to the model.

Lines 305ff: I tried to access the slopes for the images but couldn't find a respective file in the repository. Please check for completeness. Note that I did not check the python scripts for correctness or completeness.

Figure 1: Example images of the female fish would be appreciated.

Figure 2: Although I am aware that the presented images are examples, they may represent an important flaw: the vertical angle is different. On the top image one can see a "horizon line" and rays, while on the bottom image there is only ground depicted. However, since a large number of images was used to characterize the habitats, this might be no big problem. However, the authors should check, whether the angle systematically differed between habitats.

Dr. Claudia Menzel, University of Koblenz-Landau, Germany.

Referred references:

Juricevic, I., Land, L., Wilkins, A., & Webster, M. A. (2010). Visual discomfort and natural image statistics. *Perception*, 39(7), 884–899. <https://doi.org/10.1068/p6656>

Menzel, C., Hayn-Leichsenring, G. U., Langner, O., Wiese, H., & Redies, C. (2015). Fourier power spectrum characteristics of face photographs: Attractiveness perception depends on low-level image properties. *Plos One*, 10(4), e0122801. <https://doi.org/10.1371/journal.pone.0122801>

Menzel, C., Hayn-Leichsenring, G. U., Redies, C., Németh, K., & Kovács, G. (2017). When noise is beneficial for sensory encoding: Noise adaptation can improve face processing. *Brain and Cognition*, 117, 73–83. <https://doi.org/10.1016/j.bandc.2017.06.006>

Redies, C., Hänisch, J., Blickhan, M., & Denzler, J. (2007). Artists portray human faces with the Fourier statistics of complex natural scenes. *Network: Computation in Neural Systems*, 18(3), 235–248. <https://doi.org/10.1080/09548980701574496>

Wu, J., Xu, H., Dayan, P., & Qian, N. (2009). The role of background statistics in face adaptation. *The Journal of Neuroscience*, 29(39), 12035–12044. <https://doi.org/10.1523/JNEUROSCI.2346-09.2009>

Dear Editor and Reviewers,

Thank you for your time and feedback in reviewing our manuscript. We appreciate the opportunity to resubmit a revision that we feel is substantially improved by your comments. To summarize briefly, we conducted additional analyses, testing (1) whether the Fourier slope of male color patterns were more similar to their respective habitat slopes than were female patterns, and (2) whether the box size of our image analyses had an effect on our results. We also reframed the introduction and discussion to focus more explicitly on sensory drive, psychophysics, and the novelty of testing for similarity with the environment rather than contrast, as in past studies of sensory drive. We also added the references provided by the reviewers, which have strengthened the context of our study, and we tried to clarify what was unclear. We made extensive changes, rewriting most of the article, making it impossible to keep a readable copy with highlighted changes. We thus have submitted only a clean, revised copy of the manuscript. We address specific comments below, in italics.

Reviewer #1

This is an exciting finding within the field of visual ecology showing that spatial elements of sexual signals covary with habitat.

1. The data are all about the physical characteristics of signals and background, which makes it puzzling that the interpretation focuses so heavily on the subjective evaluation of signals by receivers. While the connection to empirical aesthetics (a very small field with a Max-Planck Institute) is intriguing, these results fall squarely and elegantly within the realm of visual ecology and psychophysics.

> We agree that the main framework is visual ecology and psychophysics. We re-wrote the introduction to emphasize that framework and removed some of our discussion of empirical aesthetics from the introduction, though we kept one paragraph in the end of the discussion for speculation, L209. We also added several references in the introduction as recommended: Fleishman 1992, L40; Marshall 2000, L39; Changizi 2006, L52; Rosenthal 2007, L153.

2. I understand the temptation to connect these results to the authors' recent conceptual paper on processing fluency, but I think the connection is far overstated here. The model being presented here is analogous to sensory drive: habitat characteristics drive perceptual biases. Signals are favored because they address these biases.

> We understand this concern and have addressed it with a re-write of the introduction that emphasizes sensory drive and visual ecology. We also feel that it is necessary to connect to findings in cognitive science that demonstrate human preference for efficiently coded stimuli, as these are a major impetus for hypothesizing the perceptual bias. We briefly mention these now in the introduction (L53) and expand on them in a single paragraph in the discussion L154; however, our main focus is now more squarely on sensory drive and psychophysics.

3. With these data, it is impossible to distinguish between whether signals are favored because they are easier to detect by receivers, versus whether they are processed more pleurably. I found it surprising that there wasn't more of a connection to psychophysics and what we already know about processing of spatial information. One of the interesting things about spatial frequency is that it is processed at multiple levels of the visual system, starting with receptive fields in the retina. Spatial frequency sensitivity is also famously sensitive to the early visual environment, providing a compelling mechanism for the tuning of frequency sensitivity to the local visual environment. It is possible that these spatial frequencies are more detectable or more attractive because they're parsed more easily in a given environment. But visual systems function by detecting contrast, and matching spatial frequency is a common feature of camouflage and mimicry. Indeed, I think the most parsimonious evolutionary explanation for this pattern is that the spots of male darters have evolved to minimize conspicuousness to color-insensitive fish predators. Colors thereby stand out for conspecific females while predators see a homogeneous background.

> We agree with the hypothesis that these patterns may combine camouflaged and detectable features, and we emphasize this now in the discussion, on L185. "Indeed..." We do not know of any color-insensitive predators of darters, however. Crayfish and snakes have at least two photopic opsins (LWS and SWS; Kingston & Cronin 2010, Simões et al. 2016), local fish predators like largemouth bass have similar color vision as darters (Mitchem et al. 2019), and birds are known for exceptional color vision. Nonetheless, the idea that sexual signals incorporate both cryptic and conspicuous elements is very intriguing.

4. It is worth revisiting some of the earlier literature on visual ecology, from Fleishman's work on temporal-frequency contrast in courtship displays to Changizi's work on written symbols and topological patterns in natural scenes to Justin Marshall's extensive work on communication and camouflage in reef fish. Rosenthal's Annual Review on visual communication talks about spatial frequency matching in camouflage.

> These references are now cited and provide important comparisons with and a supportive framework for our study. See; e.g., L52, L39, L153, L208.

5. Finally, this work presents an exciting analogy to auditory frequency and pattern tuning and habitat/eavesdropper effects on signal/receiver evolution. That literature, grounded in psychophysics and neurophysiology, provides more promising grounds for future research building off this data set than does the interesting, but tenuously related, field of empirical aesthetics.

> We are also intrigued by the role of efficient processing in acoustic communication. We refer to the reference by Ryan and Rand, 1990 (L39) to make the analogy, and we reduced our discussion of empirical aesthetics in the introduction and discussion. We felt that the references we added on visual psychophysics were of sufficient length that a further discussion of acoustic psychophysics would detract rather than add here.

6. The one substantive comment I would make is to reanalyze the data in terms of spatial frequency contrast and testing for contrast differences across the spatial-frequency spectrum.

This will allow you to test whether there are spatial frequencies where darter signals stand out against the background. This will then provide direct predictions for future studies on the spatial-frequency sensitivity of receivers.

> This is a good suggestion and we would like to consider it for a future study that explores the relative contribution of cryptic and detectable features in darter signals. However, in our analyses, the habitat and fish were not photographed at the same distance, so the absolute spatial frequencies do not align. The Fourier slopes, which represent how contrast changes with frequency and are ideally scale invariant, can be compared.

To summarize, this is an interesting and important study that could make a substantial impact with some scholarly revision and additional analysis.

Reviewer #2

I really enjoyed reading this manuscript. In this manuscript, the authors tested the idea that what makes a particular signal attractive is based in what the sensory system is primed to see. This idea has been around for a while in the area of human perception of art and even in the way that letters are formed. The idea is that animals are very efficient at detecting particular patterns that are common in their environment. Things like art (in case of humans) or male secondary sex characters (in the case of the darters at hand) might have elements of their color patterns that reflect this bias.

To test this, the authors took pictures of a number of male and female darter species that are widespread in their placement in the darter phylogeny. They took pictures of males and females below the second dorsal fin. They also took pictures of the environments in which the fish are found. Some are found in rocky areas, others in detritus, others over sandy substrate.

They then performed an analysis involving the fourier transform. I will admit that I do not fully understand this method

1. I am not fully clear on what the 'slope of the Fourier power spectrum' represents. This should be spelled out a little bit more clearly in the methods. Is a fourier transform? I have done these in the past, but I would have problems explaining them. Lots of folks will have never done these. So a bit of hand holding in the supplemental materials would be appreciated.

> The last paragraph of the introduction L83 now describes the slope of the Fourier power spectrum in more general terms and also links it directly to visual information processing.

2. They then performed Anovas on the slopes. They also examined the relationships between the Flourier slope of habitats and darter color patterns, but for some reason they used an MCMCglm approach. I don't understand why they didn't use a normal linear regression approach. My issue with the MCMCglm is that you usually have to specify some type of

underlying distribution. We don't know what that is in this case. I looked in the supplemental information but did not see it. The reason I am curious about this is because the confidence limits on line 133 the confidence limits for the slope gets super close to zero (1.5×10^{-17}). The value on this is pretty small (5×10^{-3}), but the upper limit on the confidence limit is 0.0252. This suggests that confidence limit is constrained to not go below zero. If this is the case, then this result needs to be reinterpreted. Again, we need more detail on the distribution of the MCMCglmm.

> *We used a generalized linear regression rather than a normal linear regression in order to model the non-independence of observations using random effects. MCMCglmm is one of the most widely used methods in phylogenetic comparative analyses, and the only available method that can simultaneously account for phylogenetic correlations in addition to other random factors (see Garamszegi, L. Z. (2014). Modern phylogenetic comparative methods and their application in evolutionary biology. Concepts Pract. London, UK. Springer.). As now detailed in L314, we used an inverse Wishart prior for the covariances and a normal prior for the fixed effects (for all effects: prior of belief parameters $\nu = 0.02$ and of variance parameter $V = 1$). Thus, the referee is right that random effects such as the phylogeny cannot be negative, contrary to fixed effects. We corrected the sentence explaining how to interpret the results, for both types of effects (L118-124). With MCMCglmm, the significance of a random effect is interpreted by comparing the DIC value of the models with and without the effect. This is now included in our supplementary information (SI L42-44).*

3. More generally, I found myself wondering whether matching the energy spectrum (i.e., congruence between the two fourier slopes) should make one attractive or whether it should make one cryptic. If we had found the opposite pattern (i.e., a good match for females, but not for males), then would we be interpreting this in the light of crypsis.

> *As above, the interplay of detectability and crypsis in sexual signaling is a fundamentally interesting question raised by these analyses. We now expand on this question in the discussion, L180-208, citing evidence that the Fourier slope is used to measure attractiveness in humans, but also to measure camouflage in other species, and speculating that female patterns may be more detectable in the breeding season due to an enlarged, white belly.*

4. I also had a bit of difficulty in interpreting figures 3 and 4.

> *We are not sure which aspects of the figure were difficult to interpret, and we welcome suggestions on how to improve their clarity.*

My recommendation is that the authors make this paper a bit more reader friendly. Many of the folks reading this will be coming from an ecology and evolution perspective, so they are going to need some hand holding on the fourier analysis section. What is a fourier transform? I have done these in the past, but I would have problems explaining them. Lots of folks will have never done these. So a bit of hand holding in the supplemental materials would be appreciated.

If the authors can fix these issues, then I think that this will make for a very nice paper in

Reviewer #3:

The manuscript “NCOMMS-19-24571 - Sexual signals of fish species mimic the spatial statistics of their habitat: evidence for processing bias in animal signal evolution” deals with the similarity of Fourier slopes between male darter fish and their habitat to test a signal drive hypothesis of evolutionary biology. It is a very well-written manuscript and a well-controlled study.

1. However, in the current state the justification of the exact research question and its novelty is not perfectly clear to me. Although I believe (but not know) that it is the first study that uses image processing methods to test a theory on sexual selection, it seems to disregard related literature from psychology, for example using human face stimuli. Therefore, I suggest the revision of the respective paragraphs in the introduction (is indicated below).

> We have now clarified the novelty of our study in the introduction; specifically, we apply the psychophysical principles of efficient coding to the evolutionary theory of sensory drive and predict that complex visual signals will match, rather than contrast, their environment. We also made reference to two face studies in the introduction, L49.

2. I am also not contented that (only) fish species are considered in this study. Analyses of other groups of species would be much appreciated to strengthen the drawn conclusions.

> We hope that our study motivates researchers of other taxonomic lineages to conduct similar investigations in a diverse suite of study organisms.

3. Furthermore, the result section needs to be revised because in the current state the analyses and their statistical indices are not clear and might be improved. Furthermore, some methodological aspects are not clear in the current state of the manuscript and should be addressed. Additionally, the discussion section doesn't deal much with the actual data presented but rather more general with the hypothesis tested. In my opinion, the current data and its limitations should be discussed in more detail. Based on the more detailed comments below, I suggest a major revision of the manuscript.

> We have indeed conducted a major revision of the manuscript, and we address these concerns as specified in the comments below.

It should be noted that I read the manuscript in the order any ‘Nature communications’ reader would read it (Abstract – Introduction – Results – Discussion – Methods – Supplemental) and, therefore, some of my comments might be overridden after reading later sections (esp. Methods). However, it is an indication that information is missing at a point where it is necessary. So the authors should revise the respective sentences, even when it is described in detail at later sections.

4. Line 31: The reference is not numbered.

> *The reference is now numbered (L29).*

5. Lines 63ff: “In parallel, other studies have shown that...”. Please cite the studies you are referring to.

> *The appropriate citations have been inserted (Graham & Redies, 2010, Redies et al., 2007a, Redies et al., 2007b, Graham & Field, 2008), see L48.*

6. Line 68: “hypothesize” hypothesized

> *This has been changed, now on L61.*

7. Line 71f: Although I agree that in most studies natural scenes were not considered in their diversity, the current sentence seems to claim that no researcher ever thought about different sceneries and habitats. I’d prefer a less absolute wording.

> *We have removed this paragraph from the introduction.*

8. Line 80ff and generally to the research question and hypothesis: For me, it is not clear whether the current study is the first attempt to test the hypothesis stated here. What is about, for example, the relation of the characteristics of human faces and their assumed natural habitat? In fact, photographs of human faces have different statistical properties than natural scenes (Redies, Hänisch, Blickhan, & Denzler, 2007).

> *Our study is different from studies of human faces in that we test a relationship between habitat variation and sexual signal variation. We are now careful to emphasize, L144-153, that the novelty of our study for evolutionary biology is predicting a match between signal and habitat, rather than contrast. To date, evolutionary biology has focused almost exclusively on detectability to explain how properties of the environment (e.g., image statistics) can influence the evolution of animal signal design; we suggest that efficient information coding may do a better job explaining the evolution of patterning in animals. We have also included a new analysis showing that, beyond the correlation, male pattern statistics are a better match to habitat statistics than females’. Thus, even though male statistics and habitat statistics are different (which is inherent in physiological and morpho-anatomical constraints or historical contingencies), this result supports our hypothesis that attractive signals may have evolved to have features resembling those of habitats (see also our response to comment 32).*

9. Further and more importantly, how does this theory/hypothesis deals with the fact that several different-looking animals live in the same habitat (even when considering only animals that use visual cues for mate attraction)?

> *This is an important point. This hypothesis does not exclude diverse sexual signals found a*

the same habitat, for multiple reasons, for example, many different patterns can match the spatial statistics of the environment; there are multiple competing selection pressures on patterns, the strength and direction of which may vary across species; and, physiological constraints or historical contingencies can prevent the evolution of a given pattern in one species but not in another. The general question of how diversity can persist despite convergent evolution has been addressed elsewhere and remains a topic of focused study; however, because it is not our topic, we did not feel it was appropriate to lengthen the discussion and speculate about possible mechanisms.

10. Additionally, the authors may also refer to other perception studies, e.g. on fore- and background segregation and sensory competition.

> We thank the reviewer for these suggestions and now include these points in the discussion, L175-179.

11. So far, I also do not understand why the properties of the entire habitat rather than particular (rewarding) elements in it should correlate with the visual cues of the fish.

> This is an interesting point that is exemplified by the octopus study now cited in the discussion (L188-190), showing that camouflaged octopuses match only key features of the environment rather than the whole scene. In our case, however, we followed standard methods from psychology (empirical aesthetics) and used the whole scene, testing whether human studies showing a correspondence between visual art and natural spatial statistics are replicable in a non-human system with attractive sexual signals.

12. I'd appreciate a more detailed explanation and justification of the research question and hypothesis. However, I should note that I am not very familiar with sensory drive models and related hypothesis. Overall, the justification of the research question and hypothesis is not entirely clear and satisfying to me.

> We appreciate the request for a more detailed explanation of the novelty and research question, which we think has made for a stronger paper. We now emphasize the novelty of testing for similarity rather than contrast with the background and how this could advance our understanding of the evolution of complex signal design.

13. Furthermore, I was surprised that fish species were selected as the target (and the only one).

> The fundamental question we are addressing is the diversity of visual patterning in sexually selected signals in nature. Many study systems would be appropriate, e.g., peacock spiders, birds of paradise, guenon monkeys, etc. Darter fish are especially appropriate given well-studied differences in habitat preferences and a diversity of visual signals. The third author has been studying these fish since 1996.

14. At one point, it should be described whether and how visual perception of fish (in water) differ from other animals (on land) and whether this has an effect on the used measure.

> *Our analyses correct habitat and fish images for darter vision, for which we previously established cone opsin sensitivities (Gumm et al. 2012). We do not have estimates of visual acuity for darters, but Caves et al., 2018 demonstrate no discontinuity in spatial acuity between land and water animals. The central hypothesis that sensory and perceptual systems adapt to local habitats and generate pattern preferences should hold across any habitat type.*

15. To allow a more universal conclusion, the authors may consider analyzing data also from other species' groups (preferably from land). Overall, the justification of the research question and hypothesis is not entirely clear and satisfying to me.

> *As above, we hope that our study motivates researchers of other taxonomic lineages to conduct similar investigations in a diverse suite of study organisms.*

16. Line 106f: The horizon line is represented by high rather than low spatial frequencies. The authors may rather refer to the change in luminance between sky and foreground for example.

> *We have changed the discussion of the Fourier transform to be more accurate L85-91.*

17. Line 111f and elsewhere: The authors may also refer to related work of other groups and research foci, such as work on visual discomfort and Fourier slope (e.g., Juricevic, Land, Wilkins, & Webster, 2010).

> *We appreciate this suggestion and now cite three references on visual discomfort in relation to the Fourier slope, including Juricevic et al. on L50 and L164.*

18. Furthermore, although I agree with the appropriateness of the Fourier slope as the selected measure, it should be considered to analyse further statistical properties (and if decided against it, it should be briefly stated why the Fourier slope is considered the only relevant measure).

> *Most studies demonstrating an effect of natural statistics on preference, sensitivity, discriminability, and discomfort in humans are based on the Fourier slope, and our aim here was to test whether these results extend to a non-human system. We are excited to pursue additional metrics and methods in future work, including an ongoing study using deep neural nets to simulate perception across multiple stages of visual processing.*

19. Line 117ff: I don't understand the indications for the statistical tests and therefore I am not able to approve their correctness. If that refers to an ANOVA, the statistical indices should be indicated correctly.

> *Thank you, the statistical tests and associated indices are now specified for all results.*

20. Further, the p-values should be indicated as $p < .001$ or similar, rather than "p = 2e-16".

> *The Nature Communication guidelines request exact p-values. We report exact p-values except in the cases where the p-value is below the floating point precision of the computer.*

21. I am also not sure why sex seems to be not included in the ANOVA as a factor. Rather, it seems that two different ANOVAs were calculated for the two sexes. The indications for the statistical tests should be revised and if appropriate the statistical analyses should be redone including all factors in one analysis.

> *After discussing with statisticians, we maintain that conducting ANOVAs and glmm on the sexes separately is more appropriate than having one model with all factors. What really interests us is whether the Fourier slope of darters varied across species, and covaried with the slope of habitats. We want to study this in males, and in females. However, we are not primarily interested in whether these correlations are influenced by the sex. Adding a sex effect (and interpreting its interaction effect, e.g., with habitat slope) would decrease the ability of the model to accurately estimate the other, more interesting effect (i.e. habitat slope). Yet, we added an additional analysis on the deviation of male and female slopes from the habitat slope, in which sex is the factor of interest (see below).*

22. Line 118ff: If “Overall, males had a higher slope than females...” refers to a main effect in the calculated ANOVA, this and the respective statistical indices should be indicated. If not (which might be indicated by “this difference was significant in 5 out of 10 species”), it shouldn’t be stated that “overall” males had a higher slope. In general, the description of the results is not precise and understandable.

> *We have shortened the summary of this comparison on L115-116, to “Overall, males had a shallower slope than females (Student t-test: $p < 2e-16$, males: -3.102 ± 0.331 SD, females: -3.268 ± 0.298 SD, $n = 576$). “*

23. Line 123: It is not clear what “phylogeny” refers here to. It is also not clear to me, why sex and species is not entered to the model as fixed factors.

> *For sex, see our response above. Similarly, in this model we are not primarily interested in estimating the species effect, and thus we did not include it as a fixed factor. We nevertheless added species as a random factor because individuals of the same species cannot be treated as independent observations. We thus added a site random factor nested within a species random factor. In addition, to account for the non-independence of observations due to shared ancestry pre-dating the species level, we added a covariance matrix estimated from the phylogenetic tree as a random effect (actually, MCMCglmm directly takes the ultrametric tree as a random variable; for details, see MCMCglmm course notes at <https://cran.r-project.org/web/packages/MCMCglmm/vignettes/CourseNotes.pdf>). In the revised manuscript, we have better explained the structure of our random effects.*

24. Lines 128ff: The authors may reconsider the numbers of decimal places.

> *We did not find any information in the Nature Communication guide to authors on the required number of decimal places for results. We would be happy to either add or remove*

decimal places if desired.

25. Line 129f: “We did not find a strong effect of the...” I’d prefer: “We found a rather small influence of...” since the authors did find a significant effect.

> Since MCMCglmm does not assign pMCMC values to random effects, the significance of random effects is typically not evaluated in this modeling framework. Instead, random effects are either included, or excluded based on whether they improve the model fit, as measured by DIC. The language used has been changed to reflect this (L118-120) and alternative models with different random effects are presented in the supplemental information (SI L42-48, Table S2a, S2b and S2c).

26. Line 135: For the sake of completeness, the statistical indices for that alternative model should be stated.

> The results for the model of female Fourier slopes has been added (L125-126).

27. Line 148ff: This paragraph might be moved to the introduction because no current data are discussed here while it rather discusses the research gap the study aims to fill. If the authors prefer to keep it in the discussion, it should be shortened and the current data should be integrated.

> We have followed this advice and developed this hypothesis and its novelty more clearly in the introduction (L33-42, L144-153) and reiterate it briefly in the first paragraph of the discussion.

28. Lines 161ff: Again, this and the following paragraph do not discuss the current findings but rather (partly redundantly) describes previous research. They should be moved to the introduction, or shortened and connected to the current data.

> We did not want to overwhelm the introduction with the details of findings from experimental psychology, so we reference them there briefly and then expand on them in the third paragraph of the discussion, in order to draw the parallel between results of experimental psychology and the results we found.

29. Line 191ff and elsewhere: Although I am much aware that it is not good manners to suggest own studies as a reviewer, I’d like to draw the authors’ attention to studies on human faces in front of patterns with different Fourier slope backgrounds (e.g., Menzel, Hayn-Leichsenring, Langner, Wiese, & Redies, 2015; Menzel, Hayn-Leichsenring, Redies, Németh, & Kovács, 2017; Wu, Xu, Dayan, & Qian, 2009). I am not asking for being cited but would like to point out that there exist studies on the perception of the similarity of statistical properties of backgrounds (which might be interpreted as ‘habitat’ in the context of perceiving and evaluating the target in the foreground) and the targets in the foreground. Furthermore, within this context, the authors may also discuss further perceptual mechanisms, such as sensory competition and (short-time) adaptation.

> *We appreciate this suggestion and have added a new paragraph in the discussion on this topic, L168-179.*

30. Line 207: Because of its importance to the current study, I'd prefer a reference to original data on the Fourier slope in terrestrial compared to aquatic environments, rather than referring to an overview book.

> *We now include a reference to Balboa and Grzywacz 2003 for this statement (see L108-109).*

31. Also for aquatic environments, it should be explicitly stated whether this difference refers to photographs made from above or below the water surface.

> *We now clarify that in Balboa and Grzywacz 2003, as in our study, images have been taken underwater (L108-109).*

32. Further, I am surprised that the absolute numbers of the Fourier slope retrieved for the habitats and fish in the current data set are not discussed in relation to the fish... ..The authors should also make clear why they think the correlation between fish and habitat slope is more informative than comparing the slopes directly.

> *We combined two comments from different places in the review that we believe make the same critical point, which is that we did not find a "match" per se between fish and habitat slopes, but rather a correlation. This point motivated us to conduct an additional analysis of how well male patterns match the absolute values of the habitat slope as compared to females. We found that male values are indeed a better match (smaller mean squared error deviation) than females' (L127-131). Studies from cognitive psychology also do not necessarily show a perfect match between the absolute slopes of artwork or aesthetic images and those of natural scenes; rather, as images approach the spatial statistics of natural scenes, they are more preferred or discriminated and less discomforting.*

33. Further, I am surprised that the absolute numbers of the Fourier slope retrieved for the habitats and fish in the current data set are not discussed in relation to the fish... and to other image categories analysed in previous studies. The slopes of the fish habitats are rather steep compared to those of natural scenes found in previous studies.

> *Our estimates of habitat slope are indeed steeper than those for terrestrial environments in the literature. We now emphasize this point on L107-108. This result is consistent with the aquatic habitat images of Balboa & Grzywacz 2003, who also found steeper slopes and suggest that turbidity in the water column could be removing high frequency detail (L108-111).*

34. Also, the slopes of the fish are even steeper (and much likely significantly different from their habitats), which is a bit surprising when looking at the images because I can see a lot of fine detail on the fish surface. I am wondering whether the curve of the power spectrum is really a straight line for the fish.

> *While the slopes of the fish are significantly steeper than those of habitat images, we do not believe this is due to nonlinearity in the fish power spectrums. We visually checked a large subset of our fish power spectrums to ensure that their log-log plots were linear (see figure 2 for an example). Although some of plots exhibited a small increase in power between 100 and 110 cycles/image, repeating the analyses using 10-100 cycles/image yielded qualitatively similar results (not shown). The steep slope of fishes may be due to relatively homogeneous texture within scales.*

35. Similar to figure 2, the authors should include example curves for the fish.

> *An example spectrum for one male is now included in Figure 2, alongside a spectrum of the corresponding habitat.*

36. Relatedly, the authors may revise their title and abstract to adjust the conclusion to their data. Since the conclusion is made based on a correlation and not a direct comparison, in my opinion, one cannot say that the fish “mimic” (title) the statistics of their habitat, and that these statistics are “compared” (abstract).

> *We have changed the title to indicate a correspondence between fish and habitat statistics, rather than a match.*

37. Discussion in general: Overall, the current results are not discussed in much detail. I suggest, revising the discussion to reduce redundancy with the introduction and to address the current data in more detail as well as the limitations of the study (e.g., in statistical terms a quite low number of habitats and fish species which are correlated; focus on only one particular group of fish species; focus only on aquatic animals; focus only on Fourier slope etc.).

> *We have substantially rewritten both the introduction and discussion, emphasizing general concepts in the introduction and adding more interpretation of our results in the discussion, as well as adding some explanation of our results in the Results section (L123, L131-133), since readers will read this section before the rationale in the Methods section. We emphasize the limitations of the study explicitly on L217-220.*

38. Line 271ff: Example images of the cropped and adapted versions of the fish photographs would be very helpful to follow the technical paragraphs and to interpret the Fourier slopes obtained. Please provide at least one example image that was used to measure the Fourier slope.

> *Figure 2 now includes examples of the cropped, processed images for both a fish and a habitat used for our analyses.*

39. Line 274: Please add a break/paragraph to separate the description for fish and habitat photographs.

> *This paragraph break has been added on L279.*

40. Line 276f: “Since the size of the habitat images is greater than the size of the darter’s flank, using a larger box size reduces variability in lower frequency coefficients.” I don’t get that sentence. Please also explain why different image size is not a problem for your analysis of Fourier slope (usually it does affect the slope). The “Additionally” at the beginning of the following sentence makes no sense to me, because the previous one didn’t explain why different image sizes are no problem.

Line 281ff: Related to the previous comment, I don’t understand why the same range of frequencies is considered for habitat and fish images which differed in size. When focusing on only a subset of frequencies for a larger image, information about the lower spatial frequencies are lost and the two slopes are not comparable. This should be explained and possibly changed.

Line 287ff: Okay, I see that the authors tried to match the analyses by adjusting the bins. However, I am not sure whether it solves the problem of differing image sizes. I do not understand why it was worked with different image sizes at all. This should be made more clear. I’d suggest identical image sizes and analysis procedures for both image types. Also, the sentence in line 286f and the following one seem to contradict each other. When only frequencies between 10 and 110 were included, how can the binning include the frequencies 10 to 200? This is not clear.

> *Based on this concern, we sampled the habitat images so they matched the size of the fish images. To ensure that our results are robust to different box sizes, we reproduced analyses at various scales. While the absolute values of the slopes for both fish and habitat did depend on the scaling parameters, this did not change the results qualitatively. These analyses are now available in Supplemental Information.*

41. Line 394ff: As stated above, it is not clear why not all relevant factors are included in one ANOVA.

> *See our response above.*

42. Line 299ff: I am not a statistician but in my understanding phylogeny should be included as a fixed rather than random factor because one might expect systematic and predictable influences of phylogeny (as for example sex which is usually entered as a fixed factor in such models). Please also explain how this information is entered in the model. So far, I cannot imagine how a phylogenetic tree can be entered to the model.

> *In phylogenetic comparative analyses, phylogeny is treated as a random variable because the model usually (as in the case of MCMCglmm) aims to estimate the effect of another variable (fixed effect) while accounting for the non-independence of observations due to shared ancestry. Thus phylogeny is not the variable of interest. As mentioned previously, MCMCglmm directly takes ultrametric phylogenetic trees to estimate the matrix of covariance between observations. For details, see Hadfield, J. D. (2010). MCMC methods for multi-response generalized linear mixed models: the MCMCglmm R package. Journal of Statistical Software, 33(2), 1-22.*

43. Lines 305ff: I tried to access the slopes for the images but couldn't find a respective file in the repository. Please check for completeness. Note that I did not check the python scripts for correctness or completeness.

> *Slopes of all images are now available in the github repository <https://github.com/svhulse/Fourier-Analysis>.*

44. Figure 1: Example images of the female fish would be appreciated.

> *Figure 2 now includes an image of a female E. caeruleum.*

45. Figure 2: Although I am aware that the presented images are examples, they may represent an important flaw: the vertical angle is different. On the top image one can see a "horizon line" and rays, while on the bottom image there is only ground depicted. However, since a large number of images was used to characterize the habitats, this might be no big problem. However, the authors should check, whether the angle systematically differed between habitats.

> *To make sure that there was no systematic bias in how we were photographing each habitat type, we measured the proportion of images in each class with a visible horizon. These proportions are now included in our supplementary information. We found no significant difference in proportions between habitat classes.*

Dr. Claudia Menzel, University of Koblenz-Landau, Germany.

Referred references:

Juricevic, I., Land, L., Wilkins, A., & Webster, M. A. (2010). Visual discomfort and natural image statistics. *Perception*, 39(7), 884–899. <https://doi.org/10.1068/p6656>

Menzel, C., Hayn-Leichsenring, G. U., Langner, O., Wiese, H., & Redies, C. (2015). Fourier power spectrum characteristics of face photographs: Attractiveness perception depends on low-level image properties. *Plos One*, 10(4), e0122801. <https://doi.org/10.1371/journal.pone.0122801>

Menzel, C., Hayn-Leichsenring, G. U., Redies, C., Németh, K., & Kovács, G. (2017). When noise is beneficial for sensory encoding: Noise adaptation can improve face processing. *Brain and Cognition*, 117, 73–83. <https://doi.org/10.1016/j.bandc.2017.06.006>

Redies, C., Hänisch, J., Blickhan, M., & Denzler, J. (2007). Artists portray human faces with the Fourier statistics of complex natural scenes. *Network: Computation in Neural Systems*, 18(3), 235–248. <https://doi.org/10.1080/09548980701574496>

Wu, J., Xu, H., Dayan, P., & Qian, N. (2009). The role of background statistics in face adaptation. *The Journal of Neuroscience*, 29(39), 12035–12044. <https://doi.org/10.1523/JNEUROSCI.2346-09.2009>

Reviewers' Comments:

Reviewer #1:

Remarks to the Author:

Let me preface this review by saying that I still think this is a wonderful data set that I think deserves to be published without needing to collect more new data. The article has been greatly improved for readability, and the authors have done an excellent job of responding to my and other reviewers' concern except for my major one, that the paper as written advocates without evidence for the role of female evaluative biases in shaping the observed pattern. I hope that my critical comments are taken as constructive, as I think the paper can become deeper and more robust with some moderate scholarly and analytical work.

The data are consistent with processing bias, which posits that stimuli are more attractive - they have higher hedonic value - BECAUSE they match the spatial-frequency tuning of females. This is testable only by measuring female neurophysiology and behavior.

If anything, there is more support for the alternative, but certainly not exclusive, prediction that spatial-frequency matching of bright colors evolves in response to natural selection by fish predators detecting prey against background. This is consistent with numerous empirical studies and with first principles of psychophysics, while the interesting hypothesis of processing bias is largely untested outside humans.

These two hypotheses are distinguishable only by examining the biology of receivers, and are by no means exhaustive. Upon rereading the ms, I was surprised not to find any discussion of body size or sexual size dimorphism. If male but not female body size scales with the spatial statistics of both male ornament and the habitat, this confounds the observed relationship: non-visual sources of selection like nest characteristics could explain variation in male body size. First-order estimates of size-habitat relationships and of conspicuousness to piscivorous fish (dichromats, not "color-insensitive" as I ineptly put it in my first review) seem like they would be easy to glean from the literature or perhaps from your group's existing data sets.

I fear that by advocating so strongly through the lens of evolutionary aesthetics, this study will join the host of papers making sweeping claims about the minds of receivers purely by focusing on signaler biology. The hypothesis of processing bias has yet to be tested for spatial frequency in darters.

In sum, this will be an outstanding, enduring empirical study if it presents a fair and agnostic discussion of alternative hypotheses, opening new ground by setting up testable hypotheses for signal evolution that rely on sensation and evaluation beyond the periphery.

Gil Rosenthal

Reviewer #2:

Remarks to the Author:

In this paper, the authors take calibrated photos of males and females from 10 different darter species and from various habitat types. They take a minimum of 50 photos for each of the sites where they collected the fish. They then ran a series of fourier transforms where they examined the power contained in the image as a function of cycles. So for each image, they calculated the slope between the power contained for each of the various cycles. The authors report a statistically significant relationship between the slopes that represent the habitat type of each species with the slopes of males, but not females. However, both of these are positive.

The interpretation of these data is that the male secondary sex characters are tuned by 'efficient information coding. If this is right, then it would be a pretty important paper to the field. However, there are a few things in here that give me pause.

First, I am confused about figure 2. Figure 2 shows a photo of a male and a female darter and then a photo of the habitat. It illustrates how the analysis considers a square photo of both the fish and the habitat. The problem is that they do not represent similar areas. The area of the box for the male darter can't be more than 2cm x 2cm. The area of the box for the habitat has to be at least 8cm x 8cm. I actually think that it might be much higher than that. This strikes me as problematic. I assume that the 'cycles' used in the fourier transform actually correspond to some type of unit. I am not an expert in the fourier transform, but my main understanding of this is that it was originally developed for sound and is often used for things which vary in time. The fourier transform converts something from change over time to change over frequencies. But the frequencies mean something. In this case, I am worried that we are comparing apples and oranges because the 'bins' from one picture do not necessarily correspond to the same scale of magnitude for another picture. Again, the picture showing the very close up pattern on the side of the rainbow darter and its similarity to the dappling pattern in the stream make me wonder about issues surrounding scale.

The other thing I wonder about is the intercept. What would it mean if the species and the habitat had similar intercepts? You can have similar slopes with wildly different intercepts. Does that happen here?

A second, related issue has to do with the resolution of the darter visual system. Eleanor Caves and others have recently published methods that focus on what types of details different animals can resolve with their visual systems. The high close-up in the picture with the darter scales makes me wonder whether the animals can see these details at all. This gets back to the issue of the scale of the pictures. The fish can probably resolve the large scale details of the habitat, but whether they can resolve the fine-scale details of the darters is unclear.

Third, I don't understand why the authors pooled the pictures into five different habitat categories as opposed to using the pictures from the individual's own habitat where it was collected. The authors have data for each individual from its own habitat where it was

collected. These should represent the best data for determining what makes for 'efficient coding'. Instead, the authors pooled the data for each of the five habitats and then looked at the correlations between the average slope for the habitat type and the slope for each individual species. This seems like it would result in a reduction in the power to detect the true relationship. I also wondered about the level at which this study is being performed. Are the slopes between the males and the habitats for individuals or for species? The interpretation (and figure 5) suggest species, but the text suggests individuals. I think that the data are sufficiently robust that the same patterns would emerge at the species level. This would get around the need to include site nested within species as a random effect.

Here's another one that I am confused on, "

Fourth, if you look at the plot between the habitat and the males, it looks like the relationship is driven by 3 species. The other 7 data points show not much of a pattern between the male and the habitat slopes. A very frequent statistical mistake is to say that two things are different because one of them has a statistically significant effect and the other does not. If you look at the 95% confidence intervals for the slopes, you can see that both the male and female 95% confidence intervals include the mean of the other. So this means that their slopes are not considered different from one another. Yes, they did a second analysis to try to get a sex difference, but we can see in the figures and in the stats for the slopes that they are both positive with one of them being more positive than the other.

Lines 117-126 talk about the correlation between the slopes, but the numbers and confidence intervals do not refer to correlations. These are slopes, not correlation coefficients.

Reviewer #3:

Remarks to the Author:

First, I'd like to thank the authors for their detailed and helpful explanations on my comments. The revised version of the manuscript is much improved and, in my opinion, suitable for publication if the following minor issues were addressed.

Line 120ff: For better readability, I'd move the last insertion of the sentence ("here, mean corresponds to effect size, pMCMC is the Bayesian equivalent of a frequentist p-value") inside the other parenthesis (i.e.: "[...] 95% CI = [0.0978, 0.437], here, mean corresponds to effect size, pMCMC is the Bayesian equivalent of a frequentist p-value; Figure 5a, Table 1), consistent with [...]"

Figure 5b: In my opinion, no regression line should be included in Figure 5b because the correlation is not significant. It might lead to a misleading conclusion if one is only checking the figures and not the text in the results section.

Discussion, esp. lines 141ff: To strengthen the main finding, the authors may empathize

that the correlation was only found for males but not females, since this sex difference only allows the conclusion on sensory drive and sexual signals.

Line 157ff: "artists' portraits of human faces mimic the Fourier slope of natural landscapes more closely than images of human faces do" -> to make the difference more clear for a naïve reader, I'd suggest "artists' portraits of human faces mimic the Fourier slope of natural landscapes more closely than PHOTOGRAPHS of human faces do"

Line 158: "Graham & Redies¹³ summarize evidence" -> "Graham & Redies¹³ summarized evidence"

Line 174: "natural-like statistics^{46,59}" -> both references are the same (Párraga et al)

Line 175ff: "Because the foreground and the background of visual scenes are not processed independently, a strong difference in their spatial statistics can generate sensory competition⁶⁰, which would further favor sexual signals that match their background." -> I don't understand this conclusion. The stronger the difference in spatial statistics the lower the sensory competition; competition should be higher the more similar the stimuli are.

Line 180/1: "the match between color pattern and habitat is better" -> since grayscale images were analysed, I'd suggest to delete "color" in that sentence

Line 194: "Lacking color," -> since the female fish are not monochrome I'd rather write "Lacking pronounced color" or similar.

Dr. Claudia Menzel, University of Koblenz-Landau, Germany

Dear Editor and Reviewers,

We appreciate your continued time and effort put into reviewing our manuscript. We believe that our paper has become stronger based on your comments, and we believe that we have addressed the reviewers' concerns, both in the manuscript and in this document. We have conducted an additional analysis to make sure body size does not account for the observed patterns. Additionally, we have verified that based on what is currently known about the darter visual system, they are able to resolve the fine details in the male patterns. We have also changed our discussion to emphasize the need for behavioral and neurophysiological data to complement this work and demonstrate a link between pattern and preference. All changes to the manuscript are highlighted in yellow. We hope that we have adequately addressed your concerns.

Reviewer #1:

1. The data are consistent with processing bias, which posits that stimuli are more attractive - they have higher hedonic value - BECAUSE they match the spatial-frequency tuning of females. This is testable only by measuring female neurophysiology and behavior. If anything, there is more support for the alternative, but certainly not exclusive, prediction that spatial-frequency matching of bright colors evolves in response to natural selection by fish predators detecting prey against background. This is consistent with numerous empirical studies and with first principles of psychophysics, while the interesting hypothesis of processing bias is largely untested outside humans.

We agree with the reviewer that the only way to demonstrate the higher hedonic value of male visual patterns would be through behavioral tests or neurophysiology. We do not intend to present these results as demonstrative, merely a result consistent with the predictions of empirical aesthetics. We have rewritten parts of the discussion to ensure this is clear, specifically by moving the paragraph on camouflage to earlier in the discussion, for emphasis, and adding the following on L219-21: "In addition, we do not yet have critical behavioral data testing whether the fish indeed prefer natural-like statistics, nor neurophysiological data testing whether these patterns are more efficiently processed."

2. These two hypotheses are distinguishable only by examining the biology of receivers, and are by no means exhaustive. Upon rereading the ms, I was surprised not to find any discussion of body size or sexual size dimorphism. If male but not female body size scales with the spatial statistics of both male ornament and the habitat, this confounds the observed relationship: non-visual sources of selection like nest characteristics could explain variation in male body size. First-order estimates of size-habitat relationships and of conspicuousness to piscivorous fish (dichromats, not "color-insensitive" as I ineptly put it in my first review) seem like they would be easy to glean from the literature or perhaps from your group's existing data sets.

One of the useful properties of the Fourier slope is that, when linear, it is scale invariant. Since it measures the rate at which contrast energy changes as scale changes, it does not depend on

the size, or extent of the image. For this reason, we do not believe that body size should have an effect on the Fourier slope, and the two factors should be independent. However, to make sure that the observed effect was not due to body size, we compared the body lengths of both males and females to their Fourier slope. We sampled five individuals for each sex and every species, and performed a Bayesian phylogenetic generalized linear model with slope as the dependent variable, size as an independent variable, species and the phylogenetic tree as random effects. The relationship between body size and Fourier slope was non-significant, for both males (mean = 0.514, pMCMC = 0.385, 95%CI = [-0.712,1.665]) and females (mean = 0.0549, pMCMC = 0.883, 95%CI = [-0.689,0.819]).

3. I fear that by advocating so strongly through the lens of evolutionary aesthetics, this study will join the host of papers making sweeping claims about the minds of receivers purely by focusing on signaler biology. The hypothesis of processing bias has yet to be tested for spatial frequency in darters.

We agree wholeheartedly. While we are enthusiastic about the merit of the evolutionary aesthetics angle, we appreciate the need to approach the data from an unbiased point-of-view. As noted above, we have moved the paragraph on camouflage to earlier in the discussion, and added the need for behavioral and neurophysiological data on L219-21.

Reviewer #2:

In this paper, the authors take calibrated photos of males and females from 10 different darter species and from various habitat types. They take a minimum of 50 photos for each of the sites where they collected the fish. They then ran a series of fourier transforms where they examined the power contained in the image as a function of cycles. So for each image, they calculated the slope between the power contained for each of the various cycles. The authors report a statistically significant relationship between the slopes that represent the habitat type of each species with the slopes of males, but not females. However, both of these are positive.

1. First, I am confused about figure 2. Figure 2 shows a photo of a male and a female darter and then a photo of the habitat. It illustrates how the analysis considers a square photo of both the fish and the habitat. The problem is that they do not represent similar areas. The area of the box for the male darter can't be more than 2cm x 2cm. The area of the box for the habitat has to be at least 8cm x 8cm. I actually think that it might be much higher than that. This strikes me as problematic. I assume that the 'cycles' used in the fourier transform actually correspond to some type of unit. I am not an expert in the fourier transform, but my main understanding of this is that it was originally developed for sound and is often used for things which vary in time. The fourier transform converts something from change over time to change over frequencies. But the frequencies mean something. In this case, I am worried that we are comparing apples and oranges because the 'bins' from one picture do not necessarily correspond to the same scale of magnitude for another picture. Again, the picture showing the very close up pattern on the side of the rainbow darter and its similarity to the dappling pattern in the stream make me wonder about issues surrounding scale.

In Figure 2, the images have been cropped and resized to make the figure more visually fluent. For the analysis, following Referee 3's previous suggestions, the boxes on both the fish and the habitats were the exact same size (200x200 pix). The boxes for the males were thus not limited to a maximal size of 2x2 cm, and those of the habitats not limited to a minimal size of 8x8 cm. The cycles used in the Fourier transform represent spatial divisions of the image. For example, for a 200x200 pix image, 2 cycles per image would correspond to a sine wave with a period of 100 pixels. As described above, one of the primary reasons for using the Fourier slope is its scale invariance. Since the slope represents the rate at which contrast changes across spatial frequencies, when the slope is linear (as shown in Figure 2, and as typically found with natural images), there is no effect of scale on the slope. Said differently, the angle of view, the level of zooming, and the distance between the camera and the object have no effect on the Fourier slope. This property has been demonstrated mathematically (Field 1993; Mumford and Gidas 2001) and empirically (e.g., Ruderman 1997; Koch et al. 2010), and is one reason why the Fourier transform is "exceptionally important in digital image processing and computer vision" (Pouli et al 2013; p123).

Field D. J. (1993). Scale-invariance and self-similar 'wavelet' transforms: an analysis of natural scenes and mammalian visual systems. in Farge M. et al. Wavelets, Fractals, and Fourier Transforms, 151-193, Clarendon press, oxford, UK.

Koch, M., Denzler, J., & Redies, C. (2010). 1/f² Characteristics and isotropy in the fourier power spectra of visual art, cartoons, comics, mangas, and different categories of photographs. PLoS one, 5(8), e12268.

Mumford, D., & Gidas, B. (2001). Stochastic models for generic images. Quarterly of Applied mathematics, 59(1), 85-111.

Pouli, T., Reinhard, E., & Cunningham, D. W. (2013). Image statistics in visual computing. AK Peters/CRC Press.

Ruderman, D. L. (1997). Origins of scaling in natural images. Vision Research, 37(23), 3385-3398.

2. The other thing I wonder about is the intercept. What would it mean if the species and the habitat had similar intercepts? You can have similar slopes with wildly different intercepts. Does that happen here?

The intercept of the Fourier slope represents the absolute luminance level in the image. Therefore, the intercept mostly reflects the lighting conditions where the pictures were taken. We did take care to ensure all images were taken in similar conditions (sunny skies between 1000 and 1500 hrs); however, luminance is not likely to affect our predictions or results, as it does not affect the slope of the Fourier power spectrum.

3. A second, related issue has to do with the resolution of the darter visual system. Eleanor Caves and others have recently published methods that focus on what types of details different animals can resolve with their visual systems. The high close-up in the picture with the darter scales makes me wonder whether the animals can see these details at all. This gets back to the issue of the scale of the pictures. The fish can probably resolve the large scale details of the habitat, but whether they can resolve the fine-scale details of the darters is unclear.

While no precise measurements of darter's visual spatial acuity have been taken, given that they are diurnal, carnivorous, and occupy shallow waters with visually complex habitats, we expect their visual acuity to be high compared to other fishes with equivalent eye sizes (approx.

3mm). From the data provided by Caves et al. (2017), the acuity of Etheostoma fishes should be above 7 cycles/degree. Additionally, as darters are pair spawners, males are physically adjacent to females during courtship, often within 5cm. At this distance, the female's ability to discern fine detail in the male pattern should exceed the resolution of the images used for our analysis. Assuming a viewing distance of 5 cm and a low darter visual acuity of 5 cycles/degree, darters can resolve fine pattern detail beyond the resolution of our images.

Caves, Eleanor M., Tracey T. Sutton, and Sönke Johnsen. (2017). Visual acuity in ray-finned fishes correlates with eye size and habitat. *Journal of Experimental Biology* 220(9): 1586-1596.

4. Third, I don't understand why the authors pooled the pictures into five different habitat categories as opposed to using the pictures from the individual's own habitat where it was collected. The authors have data for each individual from its own habitat where it was collected. These should represent the best data for determining what makes for 'efficient coding'. Instead, the authors pooled the data for each of the five habitats and then looked at the correlations between the average slope for the habitat type and the slope for each individual species. This seems like it would result in a reduction in the power to detect the true relationship.

The reviewer points out an important consideration for our study, and while we did consider analyzing each site separately, we concluded that it was most appropriate to pool our habitat images into classes. We did not match each individual with images of their capture site since the individual capture sites only represent a small portion of each individual's range, and do not fully represent the visual environment in which each species evolved. By pooling together multiple sites for each habitat class, we were better able to represent the typical visual environment of each species, removing some of the site noise. It is true that in doing so we may have lost some statistical power, but the robustness of our results to this further supports our conclusions.

5. I also wondered about the level at which this study is being performed. Are the slopes between the males and the habitats for individuals or for species? The interpretation (and figure 5) suggest species, but the text suggests individuals. I think that the data are sufficiently robust that the same patterns would emerge at the species level. This would get around the need to include site nested within species as a random effect.

The analyses were performed treating every individual fish as a separate measurement, not using the averages for each species. Figure 5 was made using the averages so that the patterns in our data could be easily visualized. Indeed, our results were qualitatively equivalent when considering each individual as a data point or when considering each species as a data point. However, treating each fish as an individual allowed us to test for a site effect, and also ensured that the interspecific variation was significant in light of intraspecific variation.

6. Fourth, if you look at the plot between the habitat and the males, it looks like the relationship is driven by 3 species. The other 7 data points show not much of a pattern between the male and the habitat slopes. A very frequent statistical mistake is to say that two things are different because one of them has a statistically significant effect and the other does not. If you look at the 95% confidence intervals for the slopes, you can see that both the male and female 95%

confidence intervals include the mean of the other. So this means that their slopes are not considered different from one another. Yes, they did a second analysis to try to get a sex difference, but we can see in the figures and in the stats for the slopes that they are both positive with one of them being more positive than the other.

For our analysis, each individual fish was treated as a separate data point, thus our results are driven by many data points. For the sex difference analysis, the numbers represent mean squared error, which is always positive. A lower MSE means there is on average less difference between the fish's slopes, and that of their habitats. In our case, the values for males were significantly lower than the values for females, indicating that males more closely match their habitats than females do.

7. Lines 117-126 talk about the correlation between the slopes, but the numbers and confidence intervals do not refer to correlations. These are slopes, not correlation coefficients.

The numbers reported in this paragraph for mean represent the mean effect size of the fixed effect. Likewise, the confidence intervals represent the 95% confidence interval in which the true effect size resides. If the confidence interval includes 0, then the fixed effect is non-significant with $\alpha = 0.05$. In the case that this was confusing, we have changed the language to refer to the relationship between the variables instead of the correlation. We agree that there are no correlation coefficients reported in these results (L118, L123).

Reviewer #3:

First, I'd like to thank the authors for their detailed and helpful explanations on my comments. The revised version of the manuscript is much improved and, in my opinion, suitable for publication if the following minor issues were addressed.

1. Line 120ff: For better readability, I'd move the last insertion of the sentence ("here, mean corresponds to effect size, pMCMC is the Bayesian equivalent of a frequentist p-value") inside the other parenthesis (i.e.: "[...] 95% CI = [0.0978, 0.437], here, mean corresponds to effect size, pMCMC is the Bayesian equivalent of a frequentist p-value; Figure 5a, Table 1), consistent with [...]"

We have made the changes suggested by the reviewer (L121).

2. Figure 5b: In my opinion, no regression line should be included in Figure 5b because the correlation is not significant. It might lead to a misleading conclusion if one is only checking the figures and not the text in the results section.

We have made the changes suggested by the reviewer.

3. Discussion, esp. lines 141ff: To strengthen the main finding, the authors may empathize that

the correlation was only found for males but not females, since this sex difference only allows the conclusion on sensory drive and sexual signals.

We have added language to the relevant sections to emphasize these results (L140).

4. Line 157ff: “artists’ portraits of human faces mimic the Fourier slope of natural landscapes more closely than images of human faces do” -> to make the difference more clear for a naïve reader, I’d suggest “artists’ portraits of human faces mimic the Fourier slope of natural landscapes more closely than PHOTOGRAPHS of human faces do”

We have made the changes suggested by the reviewer (L184).

5. Line 158: “Graham & Redies13 summarize evidence” -> “Graham & Redies13 summarized evidence”

We have made the changes suggested by the reviewer (L184).

6. Line 174: “natural-like statistics46,59” -> both references are the same (Párraga et al)

References 46 and 49 were indeed the same, and we have corrected this error, thank you.

7. Line 175ff: “Because the foreground and the background of visual scenes are not processed independently, a strong difference in their spatial statistics can generate sensory competition60, which would further favor sexual signals that match their background.” -> I don’t understand this conclusion. The stronger the difference in spatial statistics the lower the sensory competition; competition should be higher the more similar the stimuli are.

We thank the referee for stressing this inaccuracy. The literature, including the referee’s own work, has indeed revealed that similar slope values between the target and the background reduce neuronal activity, which may be explained by sensory competition, and increase both cognitive performance and attractiveness ratings. However, as stressed by the referee in one of her articles, “the mechanisms behind this positive effect [of sensory competition] remain to be further elucidated” (Menzel et al., 2017, Brain and Cognition). We changed the corresponding sentence to clarify this point in the revised manuscript (L200-205).

Menzel, C., Hayn-Leichsenring, G. U., Redies, C., Németh, K., & Kovács, G. (2017). When noise is beneficial for sensory encoding: Noise adaptation can improve face processing. Brain and Cognition, 117, 73-83.

8. Line 180/1: “the match between color pattern and habitat is better” -> since grayscale images were analysed, I’d suggest to delete “color” in that sentence

We have made the changes suggested by the reviewer (L154).

9. Line 194: "Lacking color," -> since the female fish are not monochrome I'd rather write "Lacking pronounced color" or similar.

We have made the changes suggested by the reviewer (L168)

Reviewers' Comments:

Reviewer #3:

Remarks to the Author:

The authors did a good job addressing the reviewers' comments. No further changes are needed.

Dr. Claudia Menzel